# Archaeal TFEα/β is a hybrid of TFIIE and the RNA polymerase III subcomplex hRPC62/39

Fabian Blombach[1]*, Enrico Salvadori[1,2], Thomas Fouqueau[1], Jun Yan[1†], Julia Reimann[3], Carol Sheppard[1], Katherine L Smollett[1], Sonja V Albers[3,4], Christopher WM Kay[1,2], Konstantinos Thalassinos[1], Finn Werner[1]*

[1]Institute for Structural and Molecular Biology, Division of Biosciences, University College London, London, United Kingdom; [2]London Centre for Nanotechnology, University College London, London, United Kingdom; [3]Molecular Biology of Archaea Group, Max Planck Institute for Terrestrial Microbiology, Marburg, Germany; [4]Microbiology, University of Freiburg, Freiburg, Germany

*For correspondence:
f.blombach@ucl.ac.uk (FB);
f.werner@ucl.ac.uk (FW)

Present address: [†]Department of Chemistry, University of Oxford, Oxford, United Kingdom

Competing interests: The authors declare that no competing interests exist.

**Abstract** Transcription initiation of archaeal RNA polymerase (RNAP) and eukaryotic RNAPII is assisted by conserved basal transcription factors. The eukaryotic transcription factor TFIIE consists of α and β subunits. Here we have identified and characterised the function of the TFIIEβ homologue in archaea that on the primary sequence level is related to the RNAPIII subunit hRPC39. Both archaeal TFEβ and hRPC39 harbour a cubane 4Fe-4S cluster, which is crucial for heterodimerization of TFEα/β and its engagement with the RNAP clamp. TFEα/β stabilises the preinitiation complex, enhances DNA melting, and stimulates abortive and productive transcription. These activities are strictly dependent on the β subunit and the promoter sequence. Our results suggest that archaeal TFEα/β is likely to represent the evolutionary ancestor of TFIIE-like factors in extant eukaryotes.

## Introduction

The conserved core of the archaeal and eukaryotic transcription machineries encompasses a highly complex multisubunit RNAP as well as evolutionary conserved transcription factors that govern its activities through the transcription cycle. The minimal requirements for promoter-directed and start site-specific transcription are identical for the RNAPII system in eukaryotes (*Parvin and Sharp, 1993*) and the RNAP of archaea (*Hausner et al., 1996*; *Qureshi et al., 1997*; *Werner and Weinzierl, 2002*). Two basal transcription factors, TBP (TATA-binding protein) and TFB (TFIIB in eukaryotes), are necessary and sufficient to initiate transcription in archaea in vitro. TBP and TFB facilitate promoter recognition and the recruitment of RNAP (*Werner and Grohmann, 2011*). In archaea as well as eukaryotes a third factor TFE (TFIIE) enhances the next step in initiation, the transition of the closed to the open complex (*Holstege et al., 1995*, *1996*; *Werner and Weinzierl, 2005*). Eukaryotic TFIIE is a heterodimer composed of subunits TFIIEα and TFIIEβ in humans (Tfa1 and 2 in yeast). Archaeal TFE (hereafter referred to as TFEα) is monomeric and homologous to TFIIEα, but lacks its C-terminal acidic domain (*Bell et al., 2001*; *Hanzelka et al., 2001*). To date no archaeal homologue of TFIIEβ has been identified. During open complex formation the DNA strands are separated (melted) and the template strand is loaded into the active centre of RNAP. Similar to TFIIE, TFEα facilitates open complex formation by directly interacting with the non-template DNA strand (NTS), and via an allosteric mechanism that is likely to involve structural changes in the RNAP clamp and stalk (*Grohmann et al., 2011*). While in archaea the closed-to-open complex transition occurs spontaneously (*Werner and Weinzierl, 2002*), on most eukaryotic RNAPII promoters it is dependent on the translocase activity of

**eLife digest** Life on Earth is often categorized into three domains: the eukaryotes (which include plants, animals and fungi), the bacteria and a group of unusual, single-celled microorganisms called the archaea. But several recent discoveries suggest that the origin of the eukaryotes lies within the archaeal domain. The genetic material of all of these living organisms is made up of DNA, and genes within DNA contain the instructions to make other biological molecules. Making these molecules involves first copying these instructions into a molecule of RNA via a process called transcription.

All three domains of life use enzymes called RNA polymerases (RNAPs) for transcription, and all RNAPs are thought to have originated from a common ancestor. Archaea and bacteria have a single type of RNAP, whereas all eukaryotes have at least four different kinds of RNAP. The RNAPs found in archaea share many common features with their eukaryotic counterparts. In both cases, the RNAPs do not work alone. Instead, a class of proteins known as transcription factors assist in the first step of the transcription process. One of the eukarotyic RNAPs, termed RNAP II, works with a transcription factor that contains two protein subunits (called TFIIEα and TFIIEβ). While the archaeal counterpart for TFIIEα (called TFEα) is known, the counterpart for TFIIEβ is not.

Blombach et al. have now identified the archaeal counterpart of TFIIEβ in a species of archaea called *Sulfolobus* and have renamed it TFEβ. *Sulfolobus* cells are unable to survive without this protein, which works in a similar way to TFIIEβ in assisting the RNAP to start transcription. Further analyses show that the TFEβ protein is actually related to a protein subunit that is unique to RNAP III, another eukarotyic RNAP. Both of these proteins contain clusters of iron and sulphur. Blombach et al. also found that these iron-sulphur clusters enable TFEβ to bind to its TFEα partner to form a transcription factor that can interact with the RNAP and help it to carry out transcription.

These results suggest that the TFEα/β transcription factor found in archaea is likely to resemble the ancestor of the TFIIE transcription factors found in living eukaryotes. This discovery provides new insights in the evolutionary history of both the archaeal and the eukaryotic transcription machineries.

TFIIH (*Guzman and Lis, 1999*; *Kim et al., 2000*; *Fishburn et al., 2015*). However, the dependency on TFIIH changes with DNA template topology. Strand separation and subsequent initiation from linear DNA templates strictly depends on TFIIH, but for some promoters this requirement can be overcome by using negatively supercoiled DNA templates (*Parvin and Sharp, 1993*). Under these conditions DNA melting is weakly stimulated by TFIIE, an effect that is obscured by the strong melting activity of TFIIH (*Holstege et al., 1995*). Since TFIIH is not strictly essential for RNAPII initiation per se, similar molecular mechanisms are likely to operate during open complex formation of archaeal and eukaryotic RNAPs. The archaeal model systems provide a distinct advantage in this respect allowing us to study how TFIIE/TFE facilitates DNA melting in the absence of a TFIIH-like factor.

Archaea, like bacteria, utilise one type of RNAP to execute their genetic programmes, while the transcription space of eukaryotes is partitioned into distinct and non-overlapping subsets of genes that are transcribed by 3 and 5 specialised types of RNAPs in animals and plants, respectively. The common past of all types of nuclear eukaryotic RNAP systems is reflected in the sequence, structure and function of RNAP subunits and associated basal transcription factors (*Vannini and Cramer, 2012*). TFIIE is a prominent example of this relationship. The human RNAPIII subunits hRPC62/39 (C82/34 in yeast) are positioned at the periphery of RNAPIII nearly identical to the binding site of TFIIE on RNAPII (*Vannini and Cramer, 2012*). TFEα, TFIIEα and hRPC62 contain homologous winged helix (WH) domains (*Carter and Drouin, 2010*; *Lefèvre et al., 2011*) (*Figure 1A*). TFIIEβ and hRPC39 both contain tandem WH domains suggesting their paralogous nature (*Vannini and Cramer, 2012*), although no significant sequence homology was found (*Carter and Drouin, 2010*). The origin and evolution of these factors and their role in the multiplication of parallel transcription systems in eukaryotes has remained elusive.

All crenarchaeal and most euryarchaeal genomes encode a gene that bears resemblance to the RNAPIII subunit hRPC39 (*Blombach et al., 2009*). Two regions of this gene show homology to two separate and distinct domains of eukaryotic hRPC39 proteins, namely the second of its three WH domains and a C-terminal domain that includes four highly-conserved cysteine residues (*Figure 1A*).

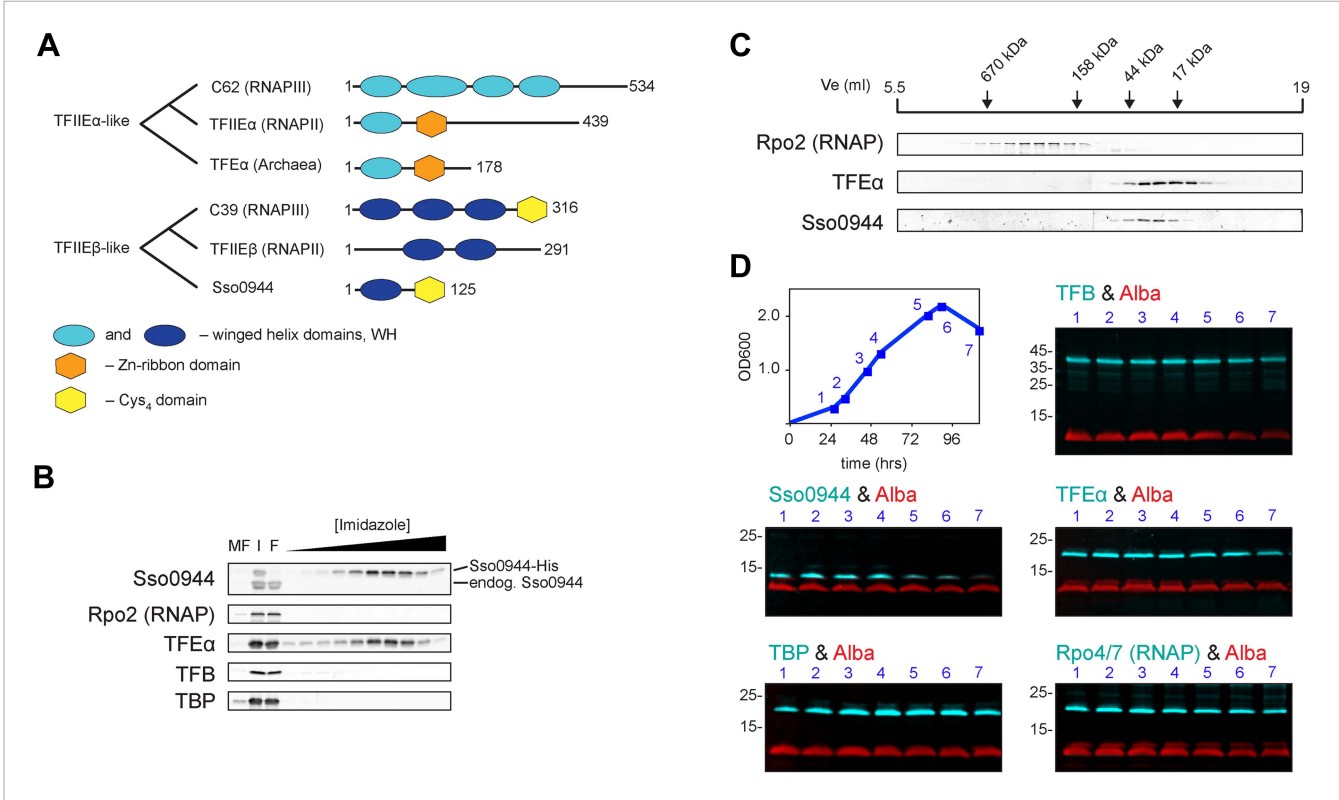

**Figure 1**. Identification of a dimeric TFE factor in *S. solfataricus*. (**A**) Conserved and acquired domains of TFIIEα and TFIIEβ-related proteins in Eukaryotes and Archaea derived from (*Blombach et al., 2009*; *Vannini and Cramer, 2012*). (**B**) Homologous expression and nickel affinity purification of Sso0944 as C-terminal His10-tag fusion. Immunodetection was used to detect co-purification of *Sso* RNAP (subunit Rpo2) and the basal transcription factors. MF—membrane fraction. I—input (soluble fraction) loaded onto column. F—flow-through fraction. (**C**) Immunodetection of RNA polymerase (RNAP), TFEα and Sso0944 (TFEβ) in *S. solfataricus* P2 cell lysate fractionated by size exclusion chromatography. (**D**) Multiplex immunodetection of *Sso* RNAP (stalk subunits Rpo4/7) and the basal transcription factors in *S. solfataricus* P2 during different growth phases. Samples were taken at the indicated time points (blue). 18 µg of lysed cells (total soluble protein content) was loaded into each lane. Immunodetection of the chromatin protein Alba served as loading control. Immunodetection of Rpo4/7 yielded a strong signal for Rpo7 at around 20 kDa, but only faint signal for Rpo4 (around 13 kDa). The experiment was carried out in triplicate and a typical result is shown.

The following figure supplements are available for figure 1:

**Figure supplement 1**. Sso0944 and TFEα form a dimeric complex.

**Figure supplement 2**. Quantitative immunodetection of TFEα, TFEβ and TBP in *S. solfataricus* P2 cell lysates during exponential growth phase (n = 3).

We have conducted a comprehensive structure-function characterisation of the archaeal hRPC39-like gene product using both in vivo and in vitro approaches. Our results show that the hRPC39-like gene from the archaeon *Sulfolobus solfataricus* is the bona fide homologue of eukaryotic TFIIEβ.

## Results

### The archaeal hRPC39-like protein is not an RNAP subunit but forms a complex with TFEα

We chose the hRPC39 homologue Sso0944 from the archaeon *S. solfataricus* (*Sso*) as model protein because the gene is a good representative of its kind (*Blombach et al., 2009*). To identify interaction partners of Sso0944 we expressed His-tagged Sso0944 in *S. solfataricus* M16 and probed the presence of co-purifying components of the basal transcription apparatus following metal-affinity

chromatography by immunodetection. While we found no evidence that the RNAP, TBP or TFB1 co-purified with Sso0944, TFEα co-eluted with Sso0944, indicating that TFEα and Sso0944 are associated in vivo (*Figure 1B*). Sypro Ruby-stained SDS-PAGE of the affinity-purified material demonstrates that the polypeptides form a dimeric complex, that is, their association is not dependent on additional factors (*Figure 1—figure supplement 1*). To rule out the possibility that the affinity tag of Sso0944 prevented its stable association with the RNAP we fractionated a wild type *S. solfataricus* P2 cell lysate by size exclusion chromatography and analysed the fractions using immunodetection. The elution profile of endogenous Sso0944 overlapped with that of TFEα consistent with a heterodimeric TFEα/Sso0944 complex of 36.1 kDa. The elution profile of TFEα was somewhat broader indicating that part of TFEα might be present in the monomeric form. RNAP eluted in earlier fractions corresponding to its molecular weight of approximately 400 kDa (*Figure 1C*). As Sso0944 is not stably incorporated into RNAP in contrast to the related RNAPIII subunit hRPC39, but rather forms a complex with TFEα, we renamed the archaeal protein TFEβ.

## TFEβ is essential for cell viability and depleted during stationary phase

Like most genes that encode basic components of the transcription apparatus, both *RPC34* and *TFA2*, the genes encoding C34 and Tfa2, respectively, are essential for cell viability in *Saccharomyces cerevisiae* (*Stettler et al., 1992*; *Feaver et al., 1994*). In order to test whether the gene encoding TFEβ is essential in *Sulfolobus* we attempted to delete the gene (*Saci_1342*) in the uracil-auxotroph *Sulfolobus acidocaldarius* strain MW001 using a pop-in/pop-out strategy and selection marker *pyrEF* (*Wagner et al., 2012*). Following genomic integration of the *Saci_1342* deletion construct, counter-selection using 5-fluoroorotic acid all clones reverted to wild type via reciprocal excision (80 clones tested) (*Table 1* and *Table 1—source data 1*). In contrast, deletion of *Saci_1342* was readily achieved in strain MW001 *Saci_1162::Saci_1342* where we introduced a second copy of *Saci_1342* replacing the non-essential gene *Saci_1162* (4 out of 20 clones) (*Table 1* and *Table 1—source data 1*). This ultimately demonstrates that the gene encoding TFEβ is essential in *S. acidocaldarius*.

In order to compare and characterise the steady-state levels of TFEα and TFEβ during exponential and stationary growth of *S. solfataricus* we carried out quantitative Western blotting. TFEα and TFEβ levels are near stoichiometric during exponential growth ($24 \pm 3$ pmol/mg soluble protein and $27 \pm 3$ pmol/mg soluble protein, respectively) and about sevenfold lower than TBP levels ($184 \pm 17$ pmol/mg) (*Figure 1—figure supplement 2*). TFEβ levels are decreased when cells enter stationary phase while RNAP, TBP, TFB and TFEα remain largely unaffected (*Figure 1D*). Our results demonstrate that dimeric TFEα/β is the predominant form of the factor in exponentially growing cells, and that the steady-state levels of the complex vary as a function of the growth cycle.

## Archaeal TFEβ and human hRPC39 contain a cubane iron-sulphur cluster

In order to carry out a structure-function analysis of TFEα/β we expressed and purified a recombinant form of the TFEα/β complex in in *Escherichia coli*. Concentrated recombinant TFEα/β has a dark brown colour and its absorption spectrum displays a shoulder at 410 nm that is characteristic for iron-sulphur cluster harbouring proteins (*Figure 2A*). In order to define the origin of this absorption more precisely we recorded continuous-wave electron paramagnetic resonance (cw-EPR) spectra. The profile of the cw-EPR spectrum with a g-value of 2.01, its sensitivity to the reducing agent sodium dithionite and the

**Table 1**. Genetic experiments showing that the TFEβ encoding gene *Saci_1342* is essential

| Parental strain | Plasmid integration relative to *Saci_1342* | Number of clones tested | Clones with *Saci_1342* deletion obtained |
|---|---|---|---|
| MW001 | upstream | 40 | 0 |
| | downstream | 40 | 0 |
| MW001 *Saci_1162::Saci_1342* | upstream | 10 | 4 |
| | downstream | 10 | 0 |

**Source data 1**. Genetic experiments showing that the TFEβ encoding gene *Saci_1342* is essential

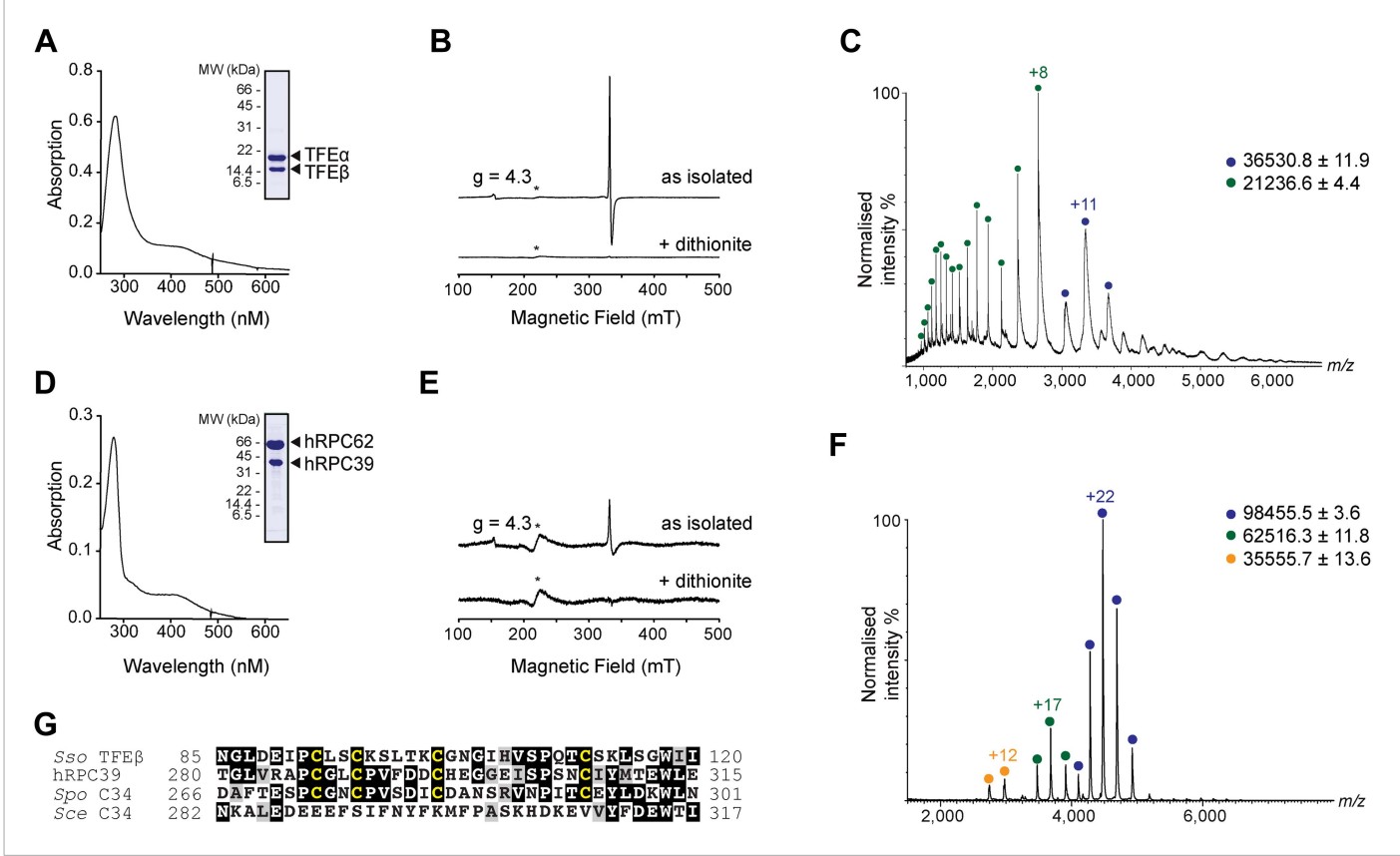

**Figure 2**. The C-terminal domain of *Sso* TFEβ harbours a 4Fe-4S cluster that is conserved in human RNAPIII subcomplex hRPC62/39. (**A**) UV-vis spectrum of TFEα/β-His. (**B**) Cw-EPR spectra of TFEα/β prepared in the presence of 1 mM DTT with or without the addition of 20 mM Na-dithionite. Besides the main [3Fe-4S]$^+$ cluster signal at g = 2.01, a small amount of spurious high spin Fe$^{3+}$ at g = 4.3 was also detected. The asterisks denote a background signal. (**C**) Nano-electrospray ionization (nESI) mass spectrum of TFEα/β-His. Filled circles indicate the different charge state series: TFEα/β-His + Zn ion + 4Fe-4S cluster (blue), TFEα + Zn ion (green). (**D**) UV-vis spectrum of recombinant human RNAP III subcomplex hRPC62/C39. (**E**) Cw-EPR spectra of hRPC62/39 prepared in the presence of the reducing agent Na-dithionite. The asterisks denote a background signal. (**F**) nESI mass spectrum of recombinant human hRPC62/C39. Filled circles indicate the different charge state series: hRPC62/C39 complex + 4Fe-4S cluster (blue), monomeric hRPC62 (green), monomeric hRPC39 (yellow). (**G**) Sequence alignment of the C-terminal domains of *S. solfataricus* TFEβ (gene id: 1455187), *H. sapiens* hRPC39 (gene id: 10621), *Schizosaccharomyces pombe* C34 (gene id: 2538992), and *S. cerevisiae* C34 (gene id: 855737). Identical and similar residues are shaded in black and grey, respectively. The four cysteine residues coordinating the FeS cluster are highlighted in yellow.

The following source data and figure supplements are available for figure 2:

**Source data 1**. Theoretical and experimentally calculated masses of proteins and protein complexes.

**Figure supplement 1**. Temperature dependence of the cw-EPR spectra of *Sso* TFEαβ and hRPC62/39.

**Figure supplement 2**. Nano-electrospray ionization mass spectrum of human hRPC62/C39 (top).

sharp decrease in signal at temperatures above 20 K are consistent with a signal corresponding to a cubane [3Fe-4S]$^+$ cluster (*Figure 2B* and *Figure 2—figure supplement 1*). Double integration and comparison with a spin standard indicate a low cluster-occupancy of approximately 4%, which is in contrast to the strong 410 nm signal in the absorption spectrum. It is common that before reduction aerobically prepared proteins containing [4Fe-4S] clusters display a [3Fe-4S]$^+$ signal due to oxidative impairment (*Beinert et al., 1996*). Addition of a reducing agent often results in the disappearance of the signal of the [3Fe-4S]$^+$ cluster concomitant with the appearance of the signal from the intact [4Fe-4S]$^+$ cluster. Here, however, only the former effect is observed: [4Fe-4S]$^{2+}$ clusters may not be reduced—and remain EPR-silent—if the redox potential is lower than that of the reducing agent

(dithionite), or if the rate of reduction is too slow. Alternatively the reduced $[4Fe-4S]^+$ cluster could be present in a high spin state, which could give rise to a signal too broad to be detected. To investigate the presence of an EPR-silent $[4Fe-4S]^+$ cluster, we recorded native mass spectra (MS) of TFEα/β (*Figure 2C* and *Figure 2—source data 1*). The spectrum contained two major species one with a mass of 21,236.6 Da and the other with a mass of 36,530.8 Da. The former corresponds to the expected mass for TFEα with a $Zn^{2+}$ ion bound by its zinc ribbon (ZR) domain while the later corresponds to the mass of the TFEα/β harbouring a $Zn^{2+}$ ion and a $[4Fe-4S]^{2+}$ cluster. Given that the TFEα ZR harbours a $Zn^{2+}$ ion, the $[4Fe-4S]^{2+}$ cluster must be coordinated by the four cysteines in the C-terminal domain of TFEβ. As this domain is highly conserved in human hRPC39 (*Figure 2G*), we examined the human protein for the presence of an equivalent FeS cluster. Similar to archaeal TFEβ, the human hRPC62/C39 complex shows an absorption spectrum with a shoulder at 410 nm (*Figure 2D*); its cw-EPR signature is characteristic for a $[3Fe-4S]^+$ cluster with low occupancy (<3%) (*Figure 2E* and *Figure 2—figure supplement 1*). The native mass spectrum this time revealed three major species with masses of 35,555.7, 62,516.3 and 98,455.5 Da corresponding to hRPC39, hRPC62 and hRPC62/C39 complex bound to a $[4Fe-4S]^{2+}$ respectively (*Figure 2F*). Tandem MS experiments revealed that the cluster was bound to hRPC39 (*Figure 2—figure supplement 2* and *Figure 2—source data 1*).

In summary, our results reveal the presence of an [4Fe-4S] cluster in the C-terminal domains of archaeal TFEβ and the human RNAPIII subunit hRPC39, thus demonstrating that this feature has been conserved through evolution. The four cysteines coordinating the [4Fe-4S] cluster show high conservation in TFEβ/hRPC39 homologs with some exceptions such as haloarchaea (*Blombach et al., 2009*) and the yeast *S. cerevisiae* C34, where all four cysteine residues have been lost (*Figure 2G*).

## The *FeS* cluster is required for TFEαβ dimerisation

TFEα and β both have a bipartite domain architectures consisting of a WH and a ZR domain, and a WH and FeS domain, respectively (*Figures 1A, 3A*). In order to characterise the interaction network between the domains in the TFEα/β heterodimer, we tested the dimerization properties of domain deletion and substitution variants using a metal affinity co-purification approach and a His-tagged TFEβ variant (*Figure 3B*). The input fractions show that all mutant variants were expressed in a soluble and heat-stable form with the exception of TFEβ Δ1–84. The TFEα WH domain is essential for dimerization (Δ1–110), while the ZR domain is dispensable (Δ114–147, *Figure 3B*). However, the WH alone is not sufficient for dimerization (Δ111–178), but requires the C-terminal tail (Δ148–178) for interaction with TFEβ. The ZR domain of *Sso* TFEα includes only three of the four conserved cysteine residues that coordinate the Zn ion in archaeal TFEα and eukaryotic TFIIEα. In line with the ZR being dispensable, mutation of the unpaired cysteine in TFEα (C117S) does not impair complex formation with TFEβ (*Figure 3B*). In yeast TFIIE, the N-terminal tip of α-helix 3 of the Tfa1 WH domain is essential for dimerization (*Grünberg et al., 2012*). We tested whether the corresponding region in the TFEα WH domain is essential for TFEα/β dimerization by introducing point mutations (K46E, D49T, R51E/K52E) or deleting the entire region (Δ46–52) (*Figure 3B* and *Figure 3—figure supplement 1*). None of these mutations abrogated dimerization with TFEβ, which reflects that the interaction network differs between yeast TFIIE and TFEα/β.

We subsequently determined the TFEβ domains required for TFEα binding. As the exact domain boundary in TFEβ is unknown we tested different deletion variants including or excluding linker residues 74–84 in dimerization experiments. The WH domain of TFEβ is not required for dimerization (Δ1–73), while the FeS domain is vital (Δ85–125, *Figure 3C*). In order to investigate the role of the FeS cluster-chelating cysteine residues in TFEβ we produced cysteine to serine mutants at positions C92, C95, C101 and C112. Following metal affinity chromatography of TFEα/β, the C92S, C95S and C112S variants rapidly lost the FeS cluster based on the absorbance spectra (*Figure 4*). Concomitant with the loss of the Fe-S cluster no complexes with TFEα were obtained. In contrast, the fourth mutation C101S had little effect on FeS cluster stability, yield and dimerization.

In summary, our results identify the TFEβ FeS domain as essential part of the TFEα/β dimerization interface that comprises also the TFEα WH and tail domains. These are sufficient to form a minimal TFEα/β complex (*Figure 3D*).

## TFEα/β binds to the RNAP clamp

*Methanocaldococcus jannaschii* TFEα binds the RNAP in a bidentate fashion, the WH domain interacts with the tip of the RNAP clamp coiled coil while the ZR domain locates to the base of the clamp and the

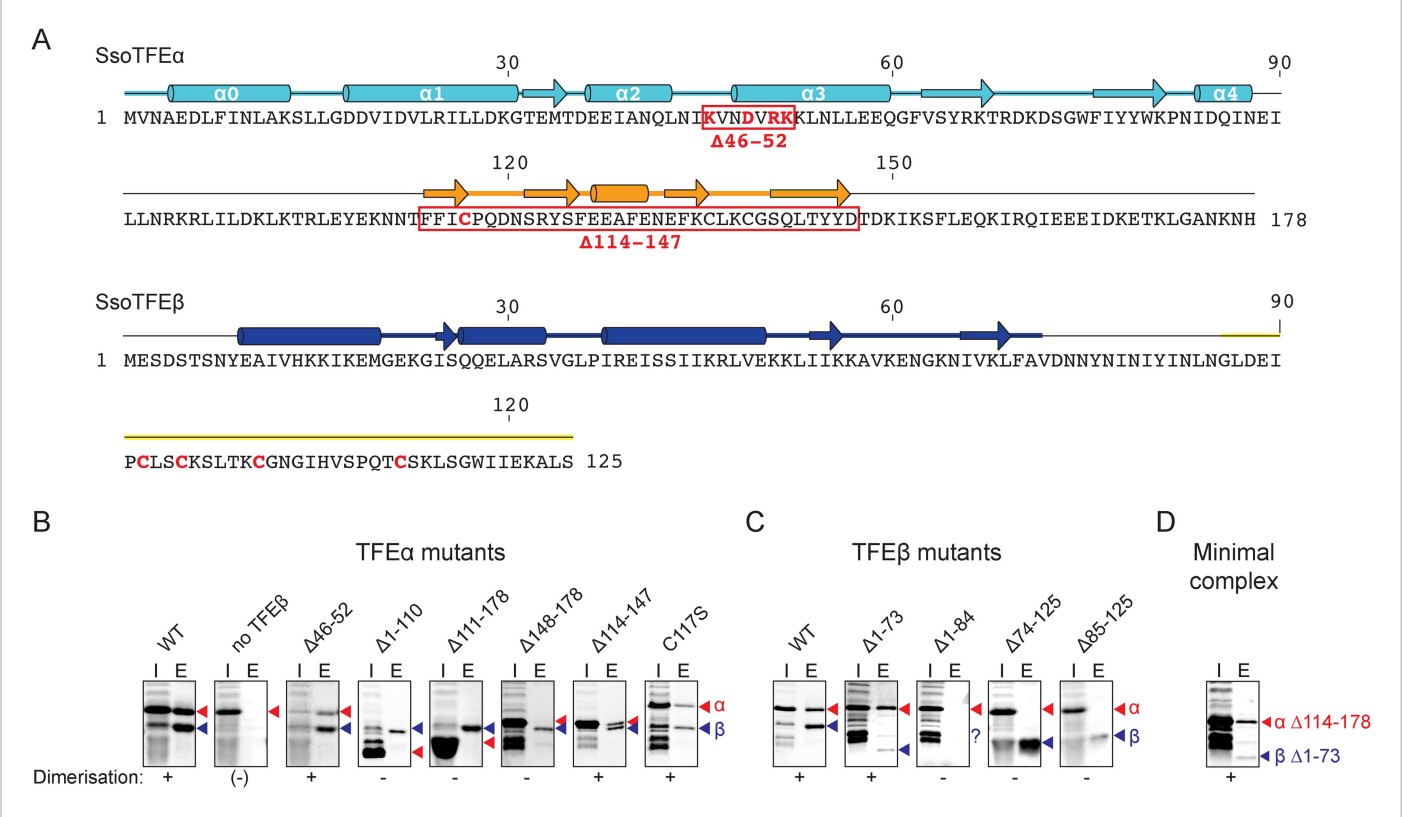

**Figure 3**. Characterisation of the TFEα/β heterodimerization interface. (**A**) Amino acid sequences and secondary structure of TFEα and TFEβ. The α-helices in the TFEα WH domain are numbered according to (*Meinhart et al., 2003*). Deletion (red boxes) and substitution deletions (highlighted in red) are indicated. (**B**, **C**) TFE α/β interaction analysis using co-purification of TFE alpha and his-tagged TFE beta. TFEα (**B**) and TFEβ-His (**C**) substitution- and domain deletion variants were co-expressed in *E. coli* and purified using Nickel-affinity chromatography. Input (heat-stable cell lysate, 25%) (I) and elution fractions (E) were analysed by SDS-PAGE and Coomassie staining. Red and blue triangles indicate the position of the respective mutant variants of TFEα and TFEβ-His, respectively. The question mark denotes that the TFEβ Δ1–84 variant is probably instable. Note that the TFEα expression levels are higher than TFEβ, and that the expression levels of TFEβ mutants are lower than WT, which in some cases make it difficult to discern in the input fractions. (**D**) Minimal heterodimeric TFEα/β complex consisting of TFEα Δ114–178 and TFEβ Δ1–73.

The following figure supplement is available for figure 3:

**Figure supplement 1**. Additional mutants of the N-terminal tip of α-helix 3 of TFEα and their effect on dimerization with TFEβ

stalk module (*Grohmann et al., 2011*). In order to characterise the binding characteristics of TFEα/β to *Sso* RNAP we produced a recombinant RNAP clamp analogously to (*Martinez-Rucobo et al., 2011*). Gel filtration elution profiles show that TFEα forms a stable complex with the recombinant RNAP clamp since both proteins eluted in earlier fractions corresponding to a larger size (*Figure 5*). The elution of both proteins was slightly asymmetrical, possibly due to partial dissociation during chromatography. Similarly dimeric TFEα/β forms a stable complex with the clamp since all three proteins co-eluted in a symmetrical fashion. Deletion of the TFEα ZR domain (TFEα ΔZR, residues 114–147) leads to loss of complex formation. In contrast, deleting the TFEβ WH domain (TFEβ ΔWH, residues 1–73) does not impair binding (*Figure 5*) which suggests that the TFEβ WH domain does not contribute to RNAP binding.

In summary, both the TFEα WH and ZR domains are required to anchor TFE to the RNAP clamp. The TFEβ FeS-domains further stabilizes the complex, whereas the TFEβ WH domain appears not to be involved.

## TFEα/β stabilizes the PIC and facilitates DNA melting

We developed an electrophoretic mobility shift assay (EMSA) to monitor the formation of the preinitiation complex (PIC) on the viral SSV1 T6 promoter (*Qureshi et al., 1997*; *Werner and*

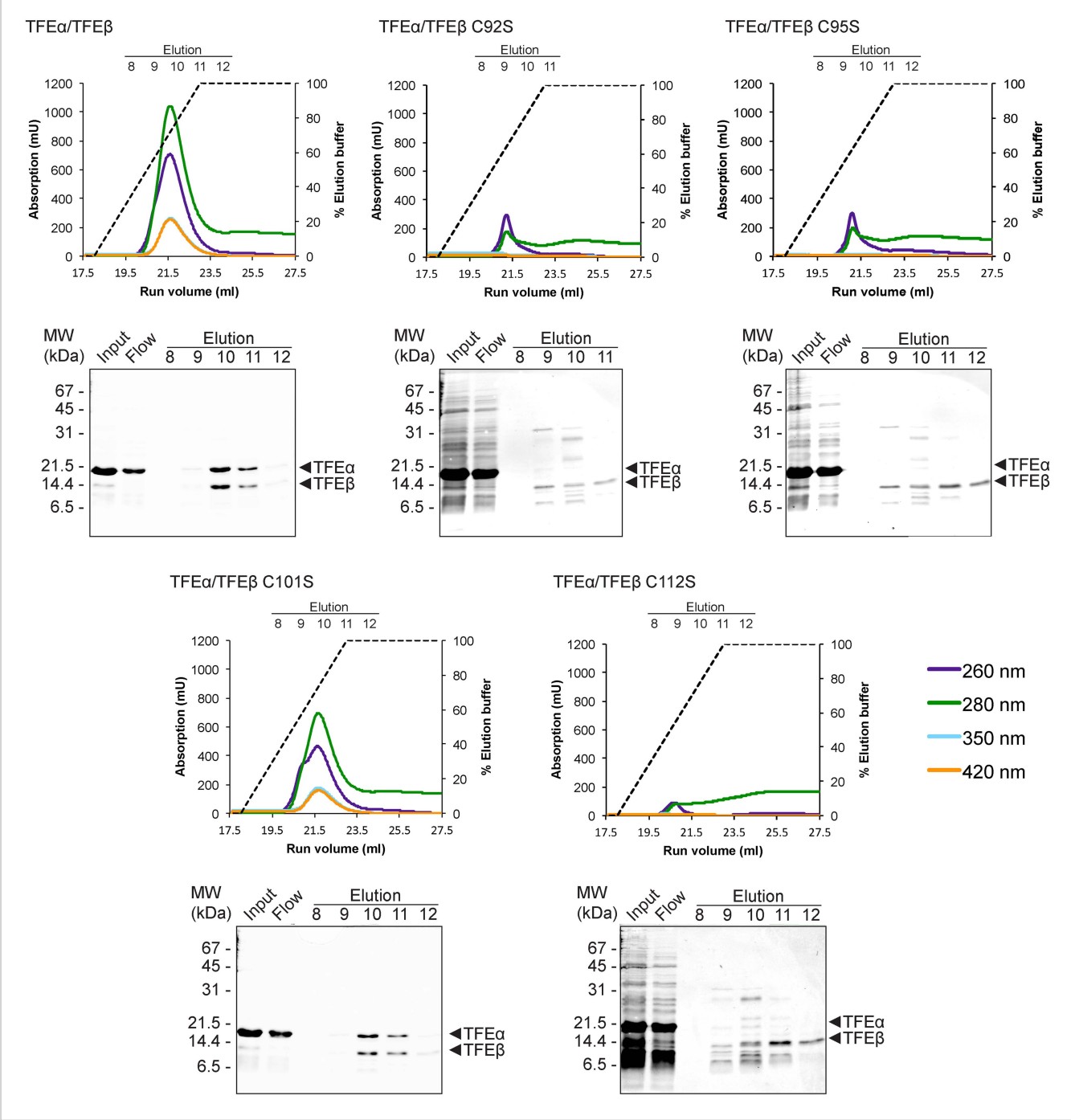

**Figure 4**. TFEαβ dimerization depends on the integrity of the Fe-S cluster. Ni-affinity chromatography of TFEβ-His (WT), TFEβ-His C92S, TFEβ-His C95S, TFEβ-His C101S or TFEβ-His C112S co-expressed with TFEα. The graphs show the absorption profile monitored at 260 nm, 280 nm, 350 nm, and 420 nm. Absorption in the visible light range (350 nm and 420 nm) is indicative for the presence of the Fe-S cluster. The elution fractions analysed by SDS-PAGE are indicated. The panels below show Coomassie-stained SDS-gels with the (heat-stable) input, flow, and elution fractions.

Weinzierl, 2005). Due to the intrinsic instability of the closed *Sso* PIC we used promoter templates that were pre-melted in the region of −4 to −1 relative to the transcription start site (TSS) (*Figure 6A*). PIC formation is strictly dependent on both TBP and TFB (*Figure 6B*). The addition of TFEα/β to the DNA-TBP-TFB-RNAP complex increases the PIC signal in a concentration-dependent fashion with the PIC now appearing in two different species indicating two different conformations. In contrast,

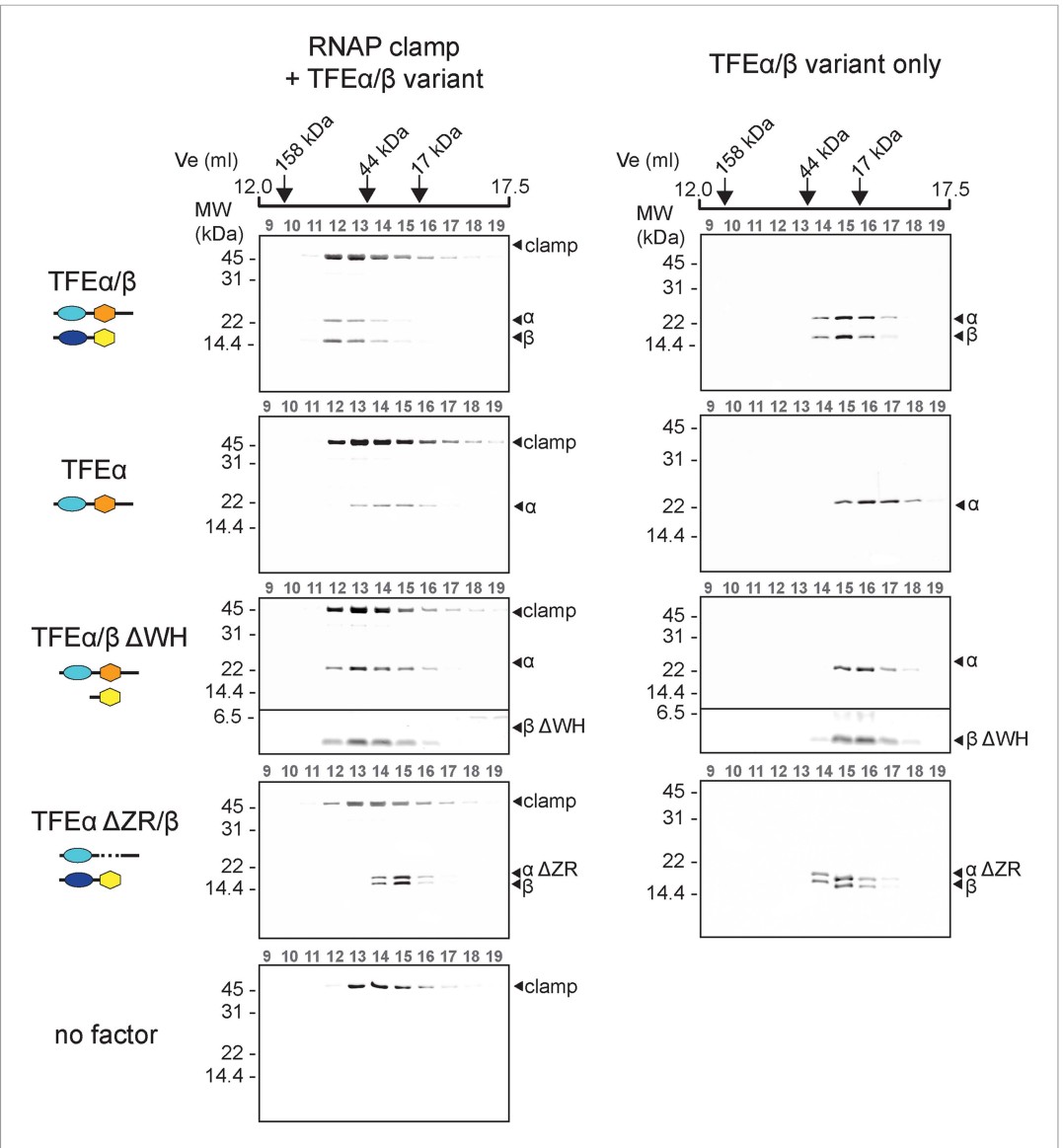

**Figure 5**. TFEα/β forms a stable interaction with the RNAP clamp module. 10 μM TFEα/β-His were incubated together with 10 μM recombinant RNAP clamp and the sample was resolved gel filtration. The presence of RNAP clamp and TFEα/β in the fractions was determined by SDS-PAGE and silver staining. The position of peaks for gel filtration marker proteins is indicated on top. For experiments with TFEα/β ΔWH the contrast was enhanced for the lower part of the gel in order to visualize TFEβ ΔWH.

monomeric TFEα had no stimulatory effect on PIC formation (*Figure 6B*). TFEα/β was not able to bind to the dsDNA template directly (*Figure 6B* and data not shown).

Deletion of the TFEα ZR or the TFEβ WH domains reduced the stimulation. The former mutation destabilizes the binding to the RNAP, while the latter variant is not impaired in RNAP binding (*Figure 5*), which suggests that the TFEβ WH domain plays a role for the stabilisation of the PIC. EMSA supershift experiments validated the incorporation of TFEβ into the PIC (*Figure 6—figure supplement 1*).

In order to test the ability of TFEα/β to facilitate DNA melting we carried out permanganate footprinting assays to identify T-residues in single-stranded regions within the NTS. The same −4/−1 pre-melted T6 promoter template was used in EMSA and permanganate assays, which accordingly result in a strong signal at −1. The ternary complex (DNA-TBP-TFB) gives in addition, and to a much lesser extent, a signal at −5 likely due to thermal breathing (*Figure 6C*). The inclusion of RNAP displayed a similar pattern, while the addition of TFEα/β led to new strong signals at −5, −7 and −12,

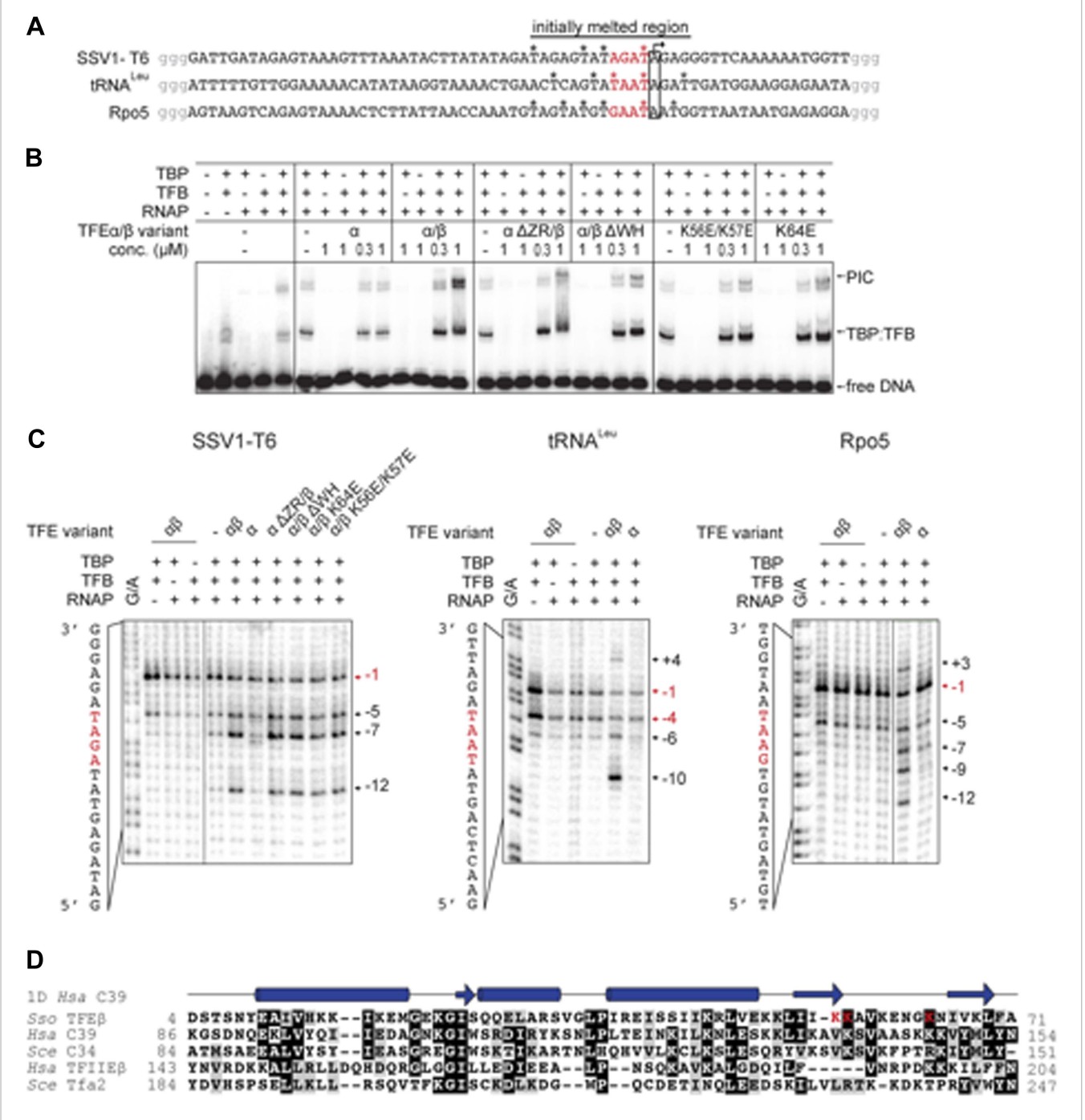

**Figure 6.** TFEα/β increases the stability of the preinitiation complex (PIC) and promotes DNA melting. (**A**) NTS sequence of the DNA templates for the promoters tested in electrophoretic mobility shift assay (EMSA) and potassium permanganate footprinting assays. The templates incude a 4 nt heteroduplex region (positions −4 to −1 relative to the TSS, in red) that was generated by introducing transition mutations into the template strand Asterisks mark permanganate-reactive T residues (see panel **C**). Additional G residues (in grey) were added to stabilize the termini. (**B**) PIC formation on the SSV1-T6 promoter in response to TFEα/β, TFEα and mutant variants using EMSAs. K56E/K57E and K64E denote mutations in the TFEβ WH. (**C**) The non-template strand of the SSV1-T6 and *Sso* Rpo5 and *Sso* tRNA[Leu] promoters was probed in potassium permanganate footprinting assays. The position of reactive T residues is indicated on the right. G/A ladder and the DNA sequence are shown on the left. (**D**) Sequence alignment of the WH domains of TFEβ (gene id: 1455187), WH2 of human C39 (gene id: 10621), WH2 of yeast C34 (gene id: 855737), WH2 of human TFIIEβ (gene id: 2961), and WH2 of yeast Tfa2 (gene id: 853936). Identical and similar residues are shaded in black and grey, respectively. On top of the alignment the secondary structure of WH2 of human C39 (PDB id: 2DK5) is depicted (α-helices as barrels and β-strands as arrows). Residues K56, K57 and K64 in the WH domain of TFEβ where mutations were introduced are highlighted in red.

*Figure 6. continued on next page*

*Figure 6. Continued*

The following figure supplement is available for figure 6:

**Figure supplement 1**. TFEα/β is part of the PIC.

which reflects that the factor stimulates open complex formation (*Figure 6C*). Deletion of the TFEβ WH or the TFEα ZR domain did not perturb the promoter opening activity of the TFEα/β complex, while TFEα was inactive, congruent with its lack of PIC stabilisation (*Figure 6C*). Similar experiments with two other −4/−1-premelted endogenous promoters from *Sso* (*Sso* Rpo5 and *Sso* tRNA^Leu) confirmed that TFEα/β strongly enhances open complex formation up to position +4 (*Figure 6C*). The WH domains of TFEβ, hRPC39/C34 and TFIIEβ/Tfa2 show little conservation, but they commonly carry sets of lysine residues in the wing formed by the two C-terminal β-strands and the loop between them (*Figure 6D*) that may play a role in electrostatic interactions with the phosphate-backbone of the NTS. We generated two charge reversal mutations in the TFEβ WH domain, K56E/K57E and K64E. In EMSA experiments, both mutants showed reduced stimulation in PIC formation when compared to WT TFEα/β (*Figure 6B*). In permanganate footprinting assays both mutants supported open complex formation similar to WT TFEα/β or TFEα/β ΔWH (*Figure 6C*).

### TFEα/β stimulates abortive and productive transcription

In order to test the influence of TFEα/β on the formation of the first phosphodiester bond we developed a promoter-dependent dinucleotide extension assay. RNAP is able to add one ATP molecule to an ApG dinucleotide to synthesize an ApGpA trinucleotide in a TBP/TFB factor-dependent fashion (*Figure 7A*). This reaction has a low TFB-independent background, but is strictly dependent on the DNA sequence of the promoter since it relies on ApG and ATP (*Figure 7A*). TFEα/β (1 μM) stimulates abortive transcription activity on closed promoter templates (2.3 ± 0.5 fold) as well as pre-opened (1.6 ± 0.1 fold) templates (*Figure 7A*) while no stimulation was observed with TFEα (data not shown).

In order to ascertain the function of TFEα/β on productive transcription we fused different promoters to a C-less cassette and carried out transcription assays in the presence of GTP, ATP and UTP. We compared the viral T6 and five cellular promoters of protein encoding genes (*Sso* EF-1α, *Sso* SSB, and *Sso* Rpo5), as well as noncoding RNA genes (*Sso* tRNA^Leu and 16S/23S rRNA) (*Figure 7B*). On all tested promoters the addition of TFEα/β stimulated transcription by approximately twofold to fourfold, with the strong T6 and 16S/23S rRNA promoters showing a weaker response (*Figure 7C* and *Figure 7—figure supplement 1*). These results suggest that the stimulation of transcription by TFEα/β is dependent on the sequence of the promoter. Considering that TFEα/β stimulates DNA melting we hypothesised that the initially melted region of the promoter (−12 to +4) determines the amplitude of the stimulation. We generated hybrid promoters encompassing the TATA-box and surrounding region (position −46 to −13) of the weakly stimulated 16S/23S rRNA promoter with the initially melted and transcribed regions of the stronger stimulated *Sso* tRNA^Leu (−12 to +5) or EF-1α promoters (−12 to +7). In absence of TFEα/β the hybrid promoters show reduced activity compared to the wild-type rRNA promoter underlining that the initially melted region contributes to the strength of the ribosomal promoter (*Figure 7D*). In line with our hypothesis TFEα/β stimulated transcription on the two hybrid promoters to greater extent when compared to the wild-type rRNA promoter, confirming that the sequence of the initially melted region determines the extent of TFEα/β stimulation. Interestingly, both TFEβ WH deletion and charge reversal mutations, and the TFEα ZR deletion mutants were able to stimulate transcription from the T6 and Rpo5 promoters (*Figure 7E* and *Figure 7—figure supplement 2*).

In summary, TFEα/β appears as a basal transcription factor that stimulates transcription on mRNA as well as noncoding RNA genes.

## Discussion

### TFEβ and hRPC39 contain structural *FeS* domains

We have discovered the TFIIEβ homologue in archaea, TFEβ, and characterised its structure and function. The gene encoding TFEβ is essential in *S. acidocaldarius* and most likely this is true for

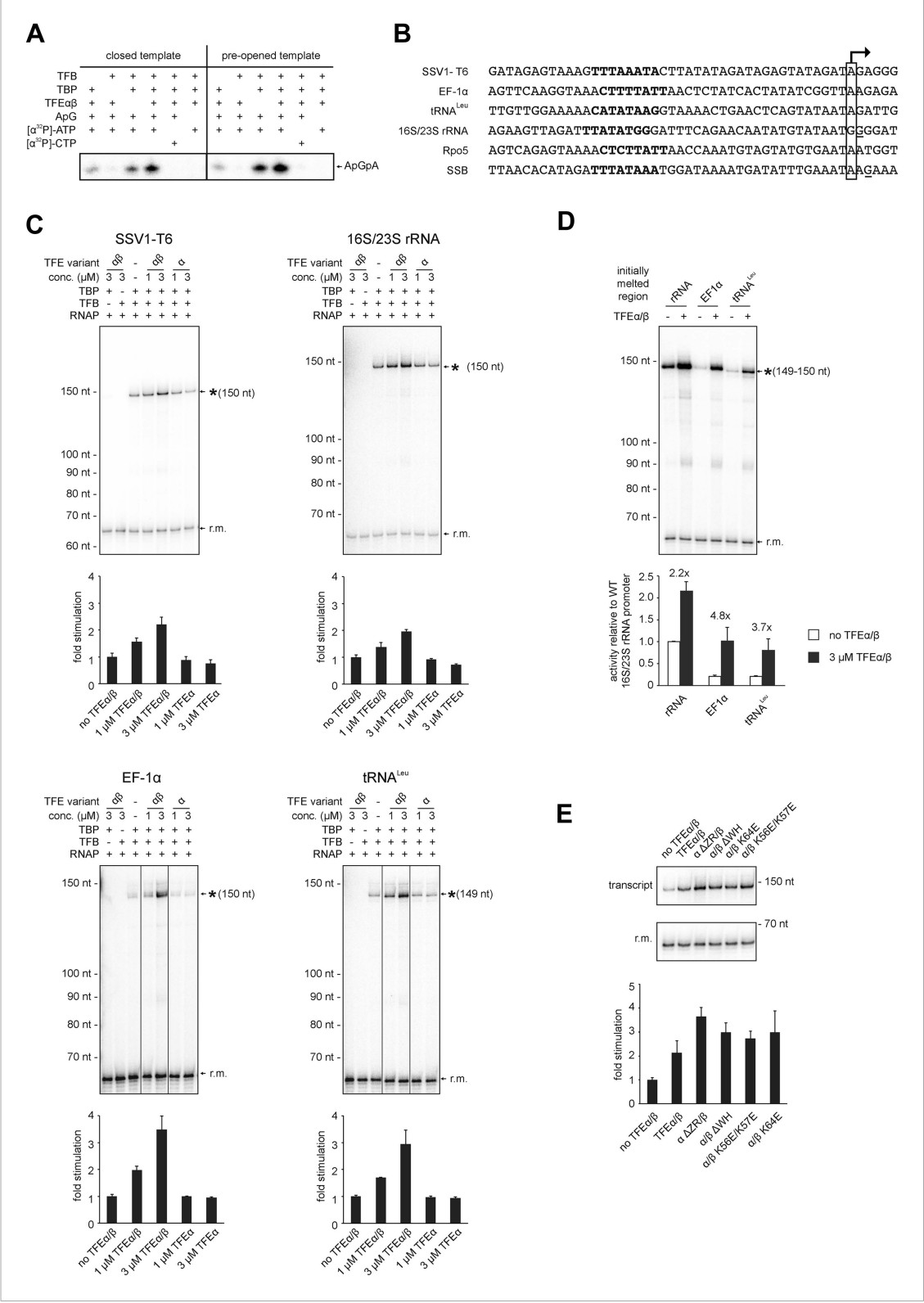

**Figure 7**. TFEα/β stimulates abortive and productive transcription. (**A**) Abortive transcription assays measuring ApGpA trinucleotide synthesis on −4/−1 heteroduplex and homoduplex SSV1-T6 promoter templates in the presence or absence of 1 μM TFEα/β. (**B**) NTS sequence of the different promoters tested in productive transcription assays. The TSS is boxed and putative TATA-boxes are shown in bold. Residues underlined are C to G mutations in order to construct a C-less cassette. (**C**) Productive transcription assays on four different promoters. Circular relaxed plasmids with different *S. solfataricus*

*Figure 7. Continued*

promoters fused to C-less cassettes were used as templates. The position of the run-off transcripts (asterisk) and its expected size are indicated. A recovery marker (r.m.) was included. The lower panels show the quantifications of synthesized transcript. (**D**) Effect of the initially melted region on TFEα/β stimulation. Productive transcription assays with hybrid promoters encompassing the TATA-box and surrounding region (position −46 to −13) of the weakly stimulated 16S/23S rRNA promoter with the initially melted and transcribed regions of the stronger stimulated tRNA$^{Leu}$ (−12 to +5) or EF-1α promoters (−12 to +7). (**E**) Effect of TFEα/β mutant variants (3 μM) on productive transcription on the T6 promoter. The mean of three technical replicates is shown. Error bars depict 1× standard deviation.

The following figure supplements are available for figure 7:

**Figure supplement 1**. Productive transcription assays using the SSB and Rpo5 promoters.

**Figure supplement 2**. Effect of TFEα/β mutant variants on productive transcription on the Rpo5 promoter.

crenarchaea in general given its strict conservation in the crenarchaeal phylum. On the sequence level TFEβ is homologous to the eukaryotic RNAPIII subunit hRPC39 and we show that both *Sso* TFEβ and human hRPC39 harbour a cubane FeS cluster at their C-termini. Several proteins involved in nucleic acid metabolism have been reported to harbour FeS clusters including multi-subunit RNAPs and TFIIH subunit XPD/Rad3 (*White and Dillingham, 2012*). Notably, the *Sso* RNAP subunit Rpo3 includes a structural [4Fe-4S] cluster that is highly stable under aerobic conditions (*Hirata et al., 2008*). Although TFEβ could be purified under aerobic conditions with high FeS cluster occupancy, the FeS cluster was lost within a few hours, as apparent from the decolouration of the protein preparation. Furthermore, oxidation with ferricyanide oxidation causes destruction of the TFEβ cluster (data not shown). This difference between the two FeS clusters in *Sso* RNAP and TFEβ likely reflects different functions. The FeS cluster in the catalytic subunit of yeast DNA Polymerase Pol δ is essential for the interaction with its two auxiliary subunits Pol31 and Pol32 (*Netz et al., 2012*). Similarly, the FeS cluster is required for TFEα/β dimerization and cluster damage results in its dissociation (*Figure 4*). Our results demonstrate that steady-state levels of TFEβ are depleted in the stationary phase—the FeS cluster in TFEα/β may provide a handle for fast inactivation of TFEα/β during stress response. In line with this hypothesis we found oxidative stress induced by hydrogen peroxide leads to rapid depletion of *Sso* TFEβ (data not shown).

## Structural organisation of TFEα/β

The minimal domain requirements for the heterodimerisation of TFEα and β encompass the TFEα WH and tail domains, and the TFEβ FeS domain (*Figures 3D, 8A*). The position of euryarchaeal TFEα on the RNAP clamp and Rpo4/7 stalk has previously been mapped in the context of the complete PIC from the euryarchaeote *M. jannaschii* (*Grohmann et al., 2011*; *Nagy et al., 2015*). Likewise, *Sso* TFEα/β forms a stable complex with a recombinant RNAP clamp. Our results show that the TFEα ZR makes extensive contacts with the RNAP clamp. The TFEβ FeS domain stabilizes the binding either by directly interacting with the RNAP clamp or indirectly by altering the conformation of TFEα (*Figure 5*). In contrast, the TFEβ WH domain is not involved. The overall structural organisation of the crenarchaeal RNAP-TFEα/β ensemble is consistent with PIC models in eukaryotes (RNAPII, *He et al., 2013*) as well as euryarchaea (*Grohmann et al., 2011*), and the architecture of RNAPIII (*Vannini et al., 2010*; *Wu et al., 2012*). The TFEα WH domain is located on the apex of the RNAP clamp, while its ZR domain projects towards and interacts with the base of the RNAP clamp and stalk module (*Grohmann et al., 2011*; *He et al., 2013*) (*Figure 8A*). The TFEβ WH domain is likely to project across the DNA binding channel prone to make contacts with the promoter template. The TFEβ FeS domain may provide additional binding surface to the RNAP clamp.

## Molecular mechanisms of TFE

Following RNAP recruitment to the promoter TFEα/β stabilises the PIC (*Figure 6B*), and this is dependent on the TFEα ZR and the TFEβ WH domain while TFEα alone has no apparent effect on PIC formation (*Figure 6B*) similar to its human homologue TFIIEα (*Peterson et al., 1991*). The proximity of *M. jannaschii* TFEα WH with the NTS at position −12 (*Grohmann et al., 2011*) and the location of the

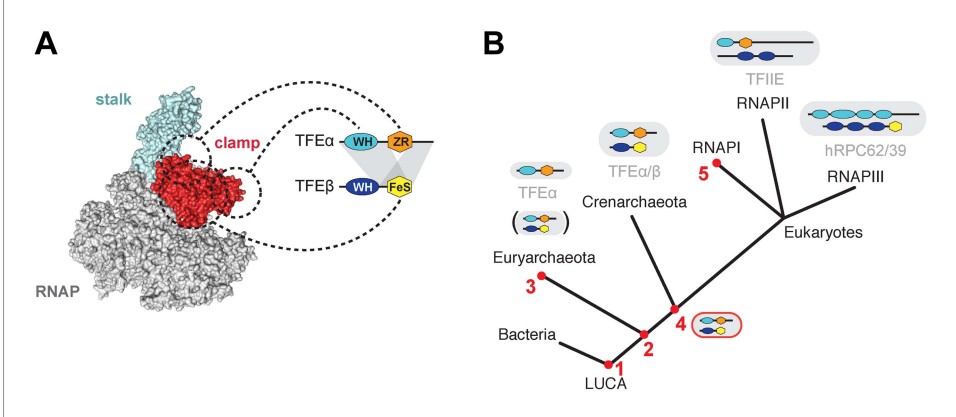

**Figure 8**. Structure and evolution of TFEα/β. (**A**) Model of the dimerization interface of TFEα/β and its interaction with the RNAP clamp based on data presented here and elsewhere (*Grohmann et al., 2011*; *Grünberg et al., 2012*; *He et al., 2013*). Grey areas indicate domains of TFEα and β that are required for dimerization. Dashed lines indicate interactions between the WH and zinc ribbon domains of TFEα and the RNAP clamp. The FeS domain of TFEβ stabilizes the interaction with the RNAP clamp and might directly bind. (**B**) A scenario for the evolution of TFIIE-like proteins in archaea and eukaryotes under assumption of an archaeal origin of eukaryotes according to the 'eocyte' hypothesis (*Guy and Ettema, 2011*; *Williams et al., 2013*). Our model includes only Euryarchaeota and Crenarchaeota, the two most studied archaeal clades. Five key steps are indicated with red numbers: 1—TFEα and TFEβ WH domains are related to bacterial MarR-type WH domains indicating their early evolutionary origin (*Aravind et al., 2005*; *Blombach et al., 2009*). 2—The wide distribution of TFEα and TFEβ encoding genes indicate that both genes date back to the last common archaeal ancestor. 3—In several euryarchaeal species the TFEβ encoding gene was lost (*Blombach et al., 2009*) and all biochemically characterised euryarchaeal TFEα appear to function as monomeric factors. Nevertheless, dimeric TFEα/β may exist in other euryarchaeal species. 4—Latest possible emergence of the dimeric TFEα/β-like factor predating the split of Crenarchaeota and the 'archaeal parent' of eukaryotes. After gene duplication, this precursor evolved to give rise to TFIIE in the RNAPII transcription machinery and the RNAPIII subcomplex hRPC62/39. 5—Eukaryotic RNAPI has lost its dependence on TFIIE-like factors.

human TFIIEβ WH across the DNA binding channel (*He et al., 2013*) suggest that TFEα/β modulates the handling of the DNA strands by RNAP, for example, during DNA melting and open complex formation. Our permanganate footprinting experiments suggest that TFEα/β, and not TFEα, triggers DNA melting creating a bubble ranging in its extremes from position +4 to −12 position relative to the TSS (*Figure 6C*). TFEα/β, but not TFEα, stimulates the formation of the first phosphodiester bond in an abortive transcription assay as well as productive transcription. The stimulatory effect of TFEα/β on DNA melting and productive transcription appears not to depend on the TFEα ZR and TFEβ WH domains (*Figures 6C, 7E* and *Figure 7—figure supplement 2*). In contrast, in experiments monitoring PIC formation using EMSA that relies to a greater extent on complex stability (as the complexes have to remain intact during electrophoresis), deletion of either TFEα ZR or TFEβ WH strongly reduces PIC stability (*Figure 6B*). Similarly, stable interaction with the RNAP clamp depends on the TFEα ZR (*Figure 5*). This suggests that the core TFEα/β composed of the TFEα WH and the TFEβ FeS domains can trigger DNA melting and thereby stimulate transcription initiation. The TFEα ZR or TFEβ WH domains bring addition stability to the complex that becomes crucial in the cellular context. In line with a stabilizing role for the ZR domain, mutation of the cysteine residues in the yeast Tfa1 ZR in vivo confer a thermosensitive phenotype (*Kuldell and Buratowski, 1997*).

We assessed the impact of TFEα/β on the transcription directed by a range of protein-encoding and noncoding RNA promoters. While hRPC39 as part of RNAPIII plays a role in the specific recruitment of RNAPIII to its noncoding RNA gene promoters, the hRPC39-like archaeal TFEβ clearly acts as a general transcription initiation factor stimulating transcription from all promoters tested. Our results show that the steady state levels of TFEβ are drastically reduced in stationary phase compared to exponential growth phase, in contrast to other components of the basal *Sso* transcription apparatus. TFEα/β stimulates transcription of different genes to varying extent, dependent on the sequence of the initiatlly melted region within the promoter. Thus, promoters that are strongly

stimulated by TFEα/β could be downregulated in stationary phase, while expression of TFEα/β unresponsive (or only mildly stimulated) promoters would be less affected. We do not provide direct evidence for a regulatory role of TFEα/β, however, our results suggest that TFEβ has the potential to reprogram transcription in response to different growth phases. TFEα/β's function as basal transcription factor and potential regulator is not unprecedented; TBP-related factors regulate transcription in metazoans (*Goodrich and Tjian, 2010*; *Duttke et al., 2014*; *Wang et al., 2014*), and multiple TBP and TFB paralogs regulate transcription in a gene-specific fashion in archaea (*Facciotti et al., 2007*).

## Evolution of TFEβ and hRPC39-like proteins in archaea and eukaryotes

The archaeal TFEα/β factor provides us with a missing link in the evolutionary history of the archaeal and multiple eukaryotic transcription machineries (*Figure 8B*). In the most parsimonious scenario, the archaeal 'parent' of eukaryotes (*Yutin et al., 2008*; *Guy and Ettema, 2011*) included a TFEα/β factor with the domain architecture we have discovered and described here. After the split of the archaeal and eukaryotic lineages about 2 billion years ago, and following gene duplication and speciation of both TFEα and β subunits in eukaryotes, the FeS domain of TFIIEβ was lost in the RNAPII system along with the evolution of a new dimer interface (*Grünberg et al., 2012*). The loss of the FeS cluster in TFIIEβ might have evolved via a route analogous to the apparent loss of the FeS cluster in yeast C34 (*Figure 2G*).

In the RNAPIII system the hRPC62/39 complex with the addition of a third, not universally conserved subunit hRPC32 (C31 in yeast) (*Proshkina et al., 2006*) became stably incorporated into RNAPIII, most likely via a WH domain duplication turning the single WH domain in TFEα to four WH domains of hRPC62 (*Lefèvre et al., 2011*) which provided a drastically enhanced binding surface (*Vannini et al., 2010*). This may have provided a selective advantage considering that transcription by RNAPIII is characterised by high rates of transcription reinitiation and short transcript length. It seems likely that the essential role of the FeS cluster in TFEα/β dimerization is conserved in hRPC62/39 since a truncated form of human hRPC39 encompassing the third WH domain and the FeS domain supports dimerization (*Lefèvre et al., 2011*).

Overall the diverse activities of TFEα/β described here are consistent with the activities of TFEα from euryarchaeal transcription systems that do not encompass TFEβ homologues such as *M. jannaschii* and *Pyrococcus furiosus* (*Hanzelka et al., 2001*; *Werner and Weinzierl, 2005*; *Naji et al., 2007*; *Kostrewa et al., 2009*; *Grohmann et al., 2011*). Monomeric *Sso* TFEα did not stimulate PIC formation or transcription (*Figures 6B, 7C*). It is worth noting that *Sso* TFEα binds to the RNAP clamp (*Figure 5*) and in vitro transcription experiments suggest that *Sso* TFEα competes with *Sso* TFEα/β for incorporation into the PIC (data not shown). Hence, *Sso* TFEα appears to be recruited to the PIC but does not exert any detectable effect on transcriptional activity. Consistent with our results, a previous study suggested that monomeric *Sso* TFEα does not affect transcription output from the viral T6 promoter when non-limiting TBP concentrations similar to those used in this study were used (*Bell et al., 2001*).

Loss of the TFEβ encoding gene in euryarchaeal species might be compensated for by changes in TFEα and RNAP, particularly regarding the mechanisms and dynamics of clamp opening. The structure of the *Thermococcus kodakaraensis* RNAP revealed the clamp in an open conformation (*Jun et al., 2014*) and this species has no TFEβ homologue. The *Sso* RNAP and yeast RNAPII crystallise with a closed clamp, and both systems utilise TF(II)Eβ for initiation (*Bushnell and Kornberg, 2003*; *Hirata et al., 2008*; *Korkhin et al., 2009*; this work). We envisage a model where TFE engages with and possibly opens the RNAP clamp, which stimulates open complex formation and transcription. In organisms that lack TFEβ the opening of the clamp occurs more readily, which is reflected in the corresponding RNAP structures; in some species clamp opening occurs entirely factor-indpendent since genes encoding TFEα and TFEβ homologues are missing in the euryarchaeal *Thermoplasmata* class. What was the selective advantage that led to the emergence of TFEβ? In budding yeast, the Tfa1 WH domain 1 interacts with the Ssl2 subunit (termed XPB in human and archaea) of TFIIH (*Grünberg et al., 2012*) which facilitates DNA nucleotide excision repair (*Rouillon and White, 2011*) and ATP-dependent DNA melting during transcription initiation (*Holstege et al., 1996*; *Kim et al., 2000*). In archaea, open complex formation occurs spontaneously due to the torsionally strained topology of the promoter DNA in the PIC (*Nagy et al., 2015*) and there is no evidence of any involvement of archaeal

XPB in transcription initiation. However, the TFEβ WH domain would be poised to enable the recruitment or integration of XPB-containing complexes into the PIC.

The discovery and characterisation of *Sso* TFEα/β provides an important piece of the evolutionary puzzle of TFIIE-like proteins. It suggests that streamlining occurred in the evolution of the archaeal transcription machinery, that is, the loss of *tfeβ* in several species (*Blombach et al., 2009*). In eukaryotes on the other hand, evolution led to an increase in complexity chiefly by WH domain duplication events.

# Materials and methods

## Molecular cloning

Cloning via restriction sites was performed with PCR products amplified from *S. solfataricus* str. P2 genomic DNA. For co-expression of *Sso* TFEα (*Sso0266*) and *Sso* TFEβ-His (*Sso 0944*) a bicistronic expression construct was produced (p1076). A corresponding construct for co-expression of hRPC62 and hRPC39 (p1159) was created likewise. To generate a pRSF-1b-based vector for expression of C-terminally His-tagged *Sso* TFEβ, the gene was amplified from a pET21a+ expression vector including the vector-encoded His-tag and inserted into the pRSF-1b vector that does not encode a His-Tag otherwise. The resulting construct (p1077) was also used as backbone to generate the different *Sso* TFEβ truncation mutants. All plasmids generated by restriction enzyme-based cloning are listed in *Supplementary file 1*. Site-directed mutagenesis of TFEα and TFEβ was performed to yield single or double amino acid substitutions as listed in *Supplementary file 2*. For the deletion of nucleotides coding for residues 46–52 in *Sso* TFEα, a 5′-end phosporylated non-overlapping primer pair was used according to the Phusion Site-directed mutagenesis protocol (Thermo Scientific/Fisher, Loughborough, United Kingdom). All oligonucleotide sequences are listed in *Supplementary file 3*. The sequence of all constructs was verified.

## Recombinant protein expression and purification

For isolation of *Sso* RNAP, pSVA158 was transformed into *S. sulfolobus* M16 cells resulting in the expression of a C-terminally His$_{10}$-tagged RNAP subunit Rpo8 and homologous expression was carried out as described previously (*Albers et al., 2006*). Cells were harvested by centrifugation, snap frozen in liquid N$_2$ and stored at −80°C. Cells were resuspended in 30 ml N buffer (25 mM Tris/HCl pH 8.0, 10 mM MgCl$_2$, 100 μM ZnSO$_4$, 5 mM 2-mercapto-ethanol, 10% glycerol) with 100 mM NaCl (N(100), salt concentration given in parenthesis) supplemented with EDTA-free protease inhibitor tablets (Roche, Burgess Hill, United Kingdom) and DNase I (Sigma, Gillingham, United Kingdom). After disruption using a French Pressure cell at 16,000 psi the lysate was cleared by centrifugation and filtration. The salt concentration was adjusted to 500 mM NaCl and the lysate was loaded onto a 1 ml Histrap ff cartridge (GE Life Sciences, Uppsala, Sweden). The column was washed with 10 ml N(500), 20 mM imidazole and RNAP was eluted using a gradient to 250 mM imidazole. Elution fractions were combined, diluted with N(0) buffer to 100 mM NaCl, and loaded onto a 1 ml HiTrap Heparin HP cartridge (GE Life Sciences). Sso RNAP was eluted with a 10 ml gradient to N(1000) yielding a single sharp peak for *Sso* RNAP. The peak fractions were combined and desalted using a PD-10 desalting column (GE Life Sciences) to N(150).

All heterologous expression was performed in *E. coli* Rosetta 2(DE3) (Merck Millipore, Billerica, MA) or BL21 Star (DE3) (Life Technologies, Paisley, United Kingdom) in enriched growth medium at 37°C according to standard procedures. For the expression of *Sso* TFEβ, *Sso* TFEα/β and all the mutant versions of these proteins as well as hC62/39 in BL21 Star (DE3) 0.5 mM L-cysteine and 0.5 mM Ammonium ferric citrate were added after induction in order to maximize FeS cluster occupancy and expression was allowed for 2.5 hr. *Sso* TBP and *Sso* TFB-His were expressed from plasmids p1121, 1087, and p1168 respectively and purified as previously described (*Gietl et al., 2014*). *Sso* TFEα was expressed in Rosetta 2(DE3) cells from plasmid p988. Cells were resuspended in N(300) and disrupted using a using a French Pressure cell at 16,000 psi. The lysate was cleared by centrifugation, incubated for 30 min at 70°C and denatured host proteins were removed by centrifugation. The heat-stable lysate was further purified on a HiPrep 16/60 Sephacryl S-100 HR column (GE Life Sciences). TFEα containing fractions were combined and threefold diluted with N(0) to 100 mM NaCl. The protein was loaded onto a UnoQ-1 column (Bio-Rad, Hemel Hempstead, United Kingdom) and eluted using a 10 ml gradient to N(1000). The purity of TFEα was judged by SDS-PAGE and fractions containing

TFEα with >95% purity were combined and concentrated by ultrafiltration, snap frozen in liquid nitrogen and stored at −80˚C. For the purification of Sso TFEα/β-His and all mutant versions, cells were resuspended in TK buffer (20 mM Tris/HCl pH 8.0, 100 µM $ZnSO_4$, 5 mM DTT) with 500 mM KCl (TK(500), salt concentration given in parenthesis) supplemented with 5 mM imidazole and disrupted by sonication. After centrifugation the supernatant was incubated at 65˚C for 20 min and centrifuged again. The heat-stable supernatant was purified on a 1 ml Histrap ff cartridge (GE Life Sciences). Sso TFEα/β-His-containing elution fractions were pooled, diluted fivefold with TK(0) buffer to 100 mM KCl and loaded on a 1 ml HiTrap Heparin HP cartridge (GE Life Sciences) and eluted with a 10 ml gradient to TK(1000). Fractions were concentrated in AMICON ULTRA 0.5 ml MWCO10000 ultrafiltration devices. Aliquots were snap-frozen in liquid nitrogen and stored at −80˚C before usage. The whole purification was carried out in a single day to limit oxidation of the Fe-S cluster. For the purification of mutant versions of Sso TFEα/β-His and WT TFEα/β-His serving as control the heparin-affinity chromatography was omitted and instead the protein was buffer changed on a PD-10 column (GE Life Sciences) to TK(150). Human N-terminally His-tagged C62/C39 was purified as follows: Cells were resuspended in TK(150), 20 mM imidazole and disrupted by sonication. After removal of cell debris by centrifugation the supernatant was purified on a 1 ml Histrap ff cartridge (GE Life Sciences). Elution fractions containing protein were combined and diluted with 0.5 vol water to 100 mM salt. The protein was further purified and buffer exchanged on a 1 ml HiTrap Heparin HP cartridge (GE Life Sciences) equilibrated in 0.1 M $NH_4$-Acetate and eluted as a single peak using a 10 ml gradient to 1 M $NH_4$-Acetate in order to have a protein preparation compatible with native MS. A construct for the expression of a recombinant Sso RNAP clamp module (p1172) was designed according to a similar construct for P. furiosus (Martinez-Rucobo et al., 2011). The Sso RNAP clamp module was purified as described above for TFEα/β-His. Protein concentrations were determined using the Qubit assay (Life Technologies).

## Western blotting and immunodetection

Polyclonal rabbit antisera against recombinant Sso TBP, Sso TFEα and Sso TFEβ were raised at Davids Biotechnology (Regensburg, Germany). For the detection of Sso TFB we used antisera raised against S. acidocaldarius TFB (Gietl et al., 2014). Rabbit antiserum against S. solfataricus RpoB was obtained from Steve Bell (Indiana University, USA) (Qureshi et al., 1997). For immunodetection, proteins were generally resolved by 14% SDS-PAGE, transferred to nitrocellulose membranes using a tank or semi-dry blotting system, and immunodetection was performed using PBS buffer with 5% milk powder as blocking reagent. The blots were incubated with the respective rabbit antisera and Dylight 680 conjugated goat anti-rabbit IgG (Thermo Scientific) as secondary antibody, scanned on a Typhoon FLA 9500 scanner (GE Life Sciences) equipped with a 685 nm laser. For multiplex immunodetection including Alba as loading control, sheep anti-Alba antiserum (obtained from Malcolm White, University of St. Andrews, UK) and donkey Dylight 488 conjugated anti-goat IgG were used alongside Dylight 680 conjugated donkey anti-rabbit IgG (Bethyl Laboratories, Cambridge, United Kingdom). We have previously shown that Alba is stably expressed throughout all growth phases (Blombach et al., 2014).

## Homologous expression and purification of Sso0944

For homologous expression of Sso0944 the gene was cloned into vector pMJ0503 (Jonuscheit et al., 2003). The resulting plasmid p1056 was transformed into S. solfataricus M16 cells and expression was carried out as described previously (Jonuscheit et al., 2003). Cells were resuspended in 20 ml TK(150) supplemented with 2.5 mM $MgCl_2$, DNase I (Sigma), and EDTA-free protease inhibitor (Roche). Cells were disrupted by threefold passage through a French pressure cell (Thermo Scientific) at 16,000 psi. Cell debris was removed by centrifugation at 30,000×g and filtration through a 0.22 µm filter. The cleared cell lysate was loaded onto a 1 ml Histrap ff cartridge (GE Life Sciences) equilibrated in TK(150) buffer and the column was washed with 5 ml TK (150) and 5 ml TK(150) containing 20 mM imidazole. Protein was eluted with a 5 ml gradient to 250 mM imidazole and 0.5 ml fractions were collected. For the more stringent purification presented in Figure 1—figure supplement 1 the second wash step was altered to 20 ml buffer containing 50 mM imidazole.

## Cell lysate fractionation

*S. solfataricus* P2 cells were grown in Brock medium, 0.1% NZ-amine, 0.2% sucrose to late exponential growth phase (OD$_{600}$ = 1.1), harvested by centrifugation and stored at −80°C. 5 g cells were resuspended in 20 ml TK(150), 10 mM MgCl$_2$, supplemented with EDTA-free protease inhibitor (Roche) and 150 µ DNase I (Sigma) and passed thrice through a French Pressure cell at 16,000 psi. The lysate was cleared by centrifugation (45,000×$g$, 1 hr at 4°C) and passage through a 0.22 µm filter. 250 µl of lysate (10 mg/ml protein content) were fractionated on a Superose 12 10/300 GL column (GE Life Sciences) with 0.5 ml fraction size. Fractions were analysed by immunodetection.

## Quantitative immunodetection

*S. solfataricus* P2 cells were grown in shake flasks in 1 litre Brock medium (*Zaparty et al., 2010*) supplemented with 0.1% (wt/vol) tryptone and 0.2% glucose at 76°C under aerobic conditions in triplicate. After 24 hr the cultures reach O.D.$_{600}$ = 0.4 (exponential growth phase) and 50 ml were withdrawn. After 96 hr at O.D.$_{600}$ = 2.3 (stationary growth phase) 10 ml were withdrawn. All samples were immediately chilled on ice, cells were harvested by centrifugation and stored at −80°C. Cells were then resuspended in 1 ml TK(150) buffer supplemented with 2.5 mM MgCl$_2$, DNase I (Sigma), and EDTA-free protease inhibitor (Roche). Cells were disrupted using a cup sonicator and lysates were cleared by centrifugation (20 min at 20,000×$g$, 4°C). Protein concentrations in the cleared lysates were determined using the Qubit assay (Life Technologies). Lysates were diluted to 1.2 mg/ml, mixed with SDS loading dye and resolved by 14% SDS-PAGE. Proteins were transferred onto nitrocellulose membranes by semi-dry blotting. To ensure equal loading and transfer, the membranes were stained with Ponceau S before immunodetection performed as described above. In parallel, a twofold dilution series of recombinant untagged TFEα/β or TBP was loaded on to the gels to generate a calibration curve used to allow determination of the expression levels.

## Saci_1342 disruption attempts

The procedure to disrupt *Saci_1342* basically followed the method described by (*Wagner et al., 2012*) using *pyrEF* as marker gene and the uracil-auxotrophic *S. acidocaldarius* strain MW001 as parental strain. The upstream and downstream flanks of *Saci_1342* (751 bp upstream flank including the initial 9 bp of *Saci_1342* and 700 bp downstream flank including 26 bp at the 3′ end of *Saci_1342*, as this region overlaps with the convergent ORF *Saci_1343*) were PCR-amplified with primer pairs FW282/283 and FW284/FW285, respectively, fused by overlap extension PCR and inserted into vector pSVA406 via NcoI and BamHI restriction sites yielding plasmid p1058. After *Esa*BC41 cytosine-methylation in *E. coli* (*Grogan, 2003*) p1058 was transformed into electrocompetent MW001 cells and single-crossover integrants were selected on uracil-deficient plates. Four single-crossover integrants were selected and plated on uracil and 5-FOA containing plates for counterselection against *pyrEF* triggering excision of p1058. Strain MW001 *Saci1162*::*Saci1342* was constructed plasmid 1112 similar to (*Meyer and Albers, 2014*). Strain MW001 *Saci1162*::*Saci_1342* was then transformed with p1058 as described above. Successful replacement of *Saci_1162* and deletion of *Saci_1342* was confirmed by DNA sequencing.

## Co-purification assays

For co-purification assays, pET-21a(+) and pRSF-1b-based vectors for the expression of *Sso* TFEα and *Sso* TFEβ or their mutant versions were co-transformed into BL21* (DE3) cells (Life Technologies). Proteins were expressed in enriched growth medium for 2.5 hr at 37°C after induction with IPTG. Cells from 400 ml culture were resuspended in 4 ml TK(500) buffer and disrupted by sonication. The cleared lysate was incubated at 65°C for 20 min. The heat-stable supernatant was mixed with 0.2 ml pre-equilibrated His-Select resin (Sigma–Aldrich) and incubated for 10 min at 4°C. The resin was washed with 5 ml buffer TK(500) containing 5 mM imidazole. Bound proteins were eluted by addition of 1 ml TK(500) containing 250 mM imidazole. Samples were analysed by Tris-Tricine SDS-PAGE.

## Binding assays with the RNAP clamp

10 µM Clamp module were mixed with 10 µM of SsoTFEαβ or mutants thereof in TK(250) buffer, 10% glycerol, 10 mM MgCl$_2$ and incubated at 37°C for 10 min. After brief centrifugation, 250 µl were loaded on a Superose 12 10/300 GL column (GE Life Sciences) equilibrated in the same buffer plus

0.05% TWEEN 20. The fraction size was 0.5 ml. The elution profile of the respective proteins was visualized on silver-stained SDS-PAGE gels.

## UV-vis spectroscopy

All samples were measured in an AvaSpec-2048 fiber optic spectrophotometer using a 10 mm path-length quartz cuvette at a protein concentration of 5 µM and 2 µM for *Sso* TFEα/β and C62/39, respectively.

## cwEPR

For EPR measurements *Sso* TFEα/β-His and human N-terminally His-tagged C62/C39 samples were aerobically purified and concentrated to 550 and 86 µM respectively, based on A280 absorption and the molar extinction coefficient. As-isolated samples were transferred directly to EPR quartz tubes and flash frozen in liquid nitrogen. Reduced samples were prepared by supplementing a fraction of the same protein stock with 20 mM sodium dithionite by adding 1/100 vol of a 2 M solution to the buffer before freezing. All samples were stored in liquid nitrogen prior to cw-EPR experiments, and showed no change in EPR signal over a period of several months. Cw-EPR measurements were performed on a Bruker EMXplus spectrometer operating at 9.4 GHz (*X*-band) equipped with a 4122SHQE resonator, with an Oxford Instruments ESR900 cryostat for measurements at cryogenic temperatures. Typically spectra were acquired in the temperature interval 10–40 K to enable FeS cluster identification. Measurements were performed with a magnetic field sweep from 50 to 600 mT (to allow the detection of possible high and low spin Fe species), a microwave power of 2 mW, modulation amplitude of 0.5 mT and a modulation frequency of 100 kHz. Three independent preparations were tested, all of which gave consistent results. Spin quantification was carried out by comparison with a standard solution of Cu(II)EDTA according to the method reported in (chasteen). The magnetic field was calibrated with a bismuth doped silicon sample.

## nESI mass spectrometry

For nano-electrospray ionization mass spectrometry of TFEα/β-His the heparin-affinity chromatography step was carried out with 0.3–1 M NH$_4$-Acetate. Prior to analysis by native mass spectrometry, protein samples were buffer-exchanged into 150 mM ammonium acetate and concentrated to ~10 µM using Amicon Ultra 0.5-ml centrifugal filters (Millipore). Mass spectrometry experiments were carried out on a first-generation Synapt HDMS (Waters, Manchester, UK) Quadrupole-TOF, traveling wave ion mobility mass spectrometer (*Pringle et al., 2007*). Samples (2- to 3-µl aliquots) were introduced to the mass spectrometer by means of nanoelectrospray (nESI) ionization using gold-coated capillaries that were prepared in-house. Typical instrumental parameters were as follows: source pressure, 5 mbar; capillary voltage, 1.0–1.3 kV; cone voltage, 40 V; trap energy, 15 V; transfer energy, 10 V; bias, 2.0 V. For tandem MS experiments, the backing pressure was reduced to 1.4 mbar and the trap and transfer voltages were increased to 60 V and 20 V respectively. MS were smoothed and peak-centered in MassLynx v4.1 (Waters, Elstree, United Kingdom).

## EMSA and potassium permanganate footprinting

15 µl samples contained 10 mM MOPS pH 6.5, 11 mM MgCl$_2$, 150 mM salt (115 mM KCl, 27 mM NaCl, 8 mM K-Acetate), 10% glycerol, 5 µg/ml heparin, 63 nM TFB, 250 nM TBP, 270 ng RNAP (45 nM) and 125 fmol of $^{32}$P 5′-labelled dsDNA templates. Samples were incubated for 5 min at 65°C before loading onto a 5% native Tris-Glycine gels (2.5% glycerol, 1 mM DTT). To generate antibody-supershifts, the Protein A-purified anti-TFEα, anti-TFEβ, or anti-TBP (0.7 µg) antiserum was added to the incubated samples and incubation continued for 15 min at room temperature prior to gel loading. Gels were run at 150 V, dried and radiolabelled DNA was detected by phosphorimagery.

For potassium permanganate footprinting the samples were as described above but scaled up to 23 µl. TBP and TFB concentrations were raised to 1 µM and 125 nM, respectively, MgCl$_2$ to 26 mM, and DTT was reduced to 0.5 mM. Samples were incubated for 5 min at 65°C before the addition of 2 µl 50 mM KMnO$_4$ (4 mM final concentration). Samples were further incubated at 65°C for 5 min before the addition of 1.5 µl 2-mercaptoethanol to stop the reaction. After Proteinase K-treatment and ethanol-precipitation, samples were resuspended in 50 µl 1 M piperidine and incubated at 90°C for 30 min. After chloroform-extraction, samples were ethanol-precipitated and resuspended in formamide loading dye.

Samples were resolved on a 10% polyacrylamide, 7 M Urea, 1× TBE sequencing gels. Radiolabelled DNA was detected by phosphor imagery.

## Abortive and productive transcription assays

15 µl samples for abortive transcription assays contained 500 fmol of dsDNA template pol592/593 (homoduplex) or pol592/603 (−4 to −1 heteroduplex), 250 µM ApG dinucleotide, 50 µM ATP (containing [$\alpha$-$^{32}$P]-ATP), 125 nM TFB, 1 µM TBP, 270 ng RNAP (45 nM), and 1 µM TFE$\alpha/\beta$. Salt, buffer and heparin concentrations were identical to those used for the mobility shift experiments. Samples were incubated for 10 min at 65°C before addition of 1 vol formamide loading dye. 5 µl of the samples was loaded on a 20% 7 M Urea, 1× TBE PAGE mini-gel and resolved at 300 V for 45 min. Signals were detected by phosphorimagery and quantified using the ImageQuant TL software package (GE Life Sciences). The quantification of TFE$\alpha/\beta$ stimulation was based on three technical replicates. The signal obtained in absence of TFB was subtracted as background.

For promoter-directed in vitro transcription, different promoters fused to a C-less cassette derived from a synthetic 390 nt G-less cassette (kindly provided by James Goodrich, CO) (*Sawadogo and Roeder, 1985*) were cloned into pGEM-T (Promega) (*Supplementary file 4*). All plasmids were treated converted to relaxed topology with *E. coli* Topoisomerase I (NEB) according to manufacturer's protocol. Transcription reactions were modified as follows: samples contained 500 µM ATP/GTP, 2.5 µM UTP (containing [$\alpha$-$^{32}$P]-UTP) and 200 ng of the respective plasmid template. Reactions were stopped after 10 min by the addition of 9 vol stop mix (0.3 M Na-acetate pH 5.2, 10 mM EDTA, 0.5% SDS, 100 µg/ml glycogen and trace amounts of radiolabelled oligonucleotide serving as recovery marker). Samples were purified by phenol/chloroform extraction and ethanol-precipitation and resuspended in formamide-loading dye. Sample were resolved on a 10% polyacrylamide, 7 M Urea, 1× TBE sequencing gel. Transcripts were detected by phosphor imagery and quantification of bands was performed using the ImageQuant TL software (GE Life Sciences). The signals were normalized using the recovery marker. When activities on different promoters were compared, the different specific activities of the transcripts were taken into account.

## Acknowledgements

The authors wish to thank Alessandro Vannini from the Institute for Cancer Research for the gift of the human hRPC62/39 expression vector, Peter Rich (UCL ISMB) for help with UV-Vis spectroscopy, and Richard Cammack for stimulating discussions of cubane FeS cluster EPR spectra. This work was supported by a Wellcome Trust investigator award to FW (079351/Z/06/Z), and FB was supported by a research fellowship from the German Research Foundation (DFG BL 1189/1-1). JR and SVA were supported by intramural fund of the Max Planck Society. JY was supported by a Wellcome Trust PhD studentship.

## Additional information

### Funding

| Funder | Grant reference | Author |
| --- | --- | --- |
| Wellcome Trust | 079351/Z/06/Z | Finn Werner |
| Deutsche Forschungsgemeinschaft | Research Fellowship DFG BL 1189/1-1 | Fabian Blombach |
| Max-Planck-Gesellschaft | intramural fund | Julia Reimann, Sonja V Albers |
| Wellcome Trust | PhD fellowship | Jun Yan |
| Biotechnology and Biological Sciences Research Council (BBSRC) | BB/H019332/1 | Finn Werner |

The funders had no role in study design, data collection and interpretation, or the decision to submit the work for publication.

## Author contributions
FB, KT, Conception and design, Acquisition of data, Analysis and interpretation of data, Drafting or revising the article; ES, Acquisition of data, Analysis and interpretation of data, Drafting or revising the article; TF, Conception and design, Acquisition of data, Analysis and interpretation of data; JY, Acquisition of data, Analysis and interpretation of data; JR, Conception and design, Acquisition of data; CS, KLS, Conception and design, Analysis and interpretation of data; SVA, Conception and design, Contributed unpublished essential data or reagents; CWMK, FW, Conception and design, Analysis and interpretation of data, Drafting or revising the article

## Author ORCIDs
Christopher WM Kay, http://orcid.org/0000-0002-5200-6004

## Additional files

### Supplementary files
• Supplementary file 1. List of plasmids generated by restriction enzyme-based cloning.

• Supplementary file 2. List of plasmids generated by PCR-based Site-directed mutagenesis.

• Supplementary file 3. List of oligonucleotides used in this study.

• Supplementary file 4. List of in vitro transcription templates with promoters fused to a C-less cassette.

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
