## [Decision Letter]

Thank you for submitting your work entitled “Archaeal TFEα/β is a hybrid of TFIIE and the RNA polymerase III subcomplex hRPC62/39” for peer review at *eLife*. Your submission has been favorably evaluated by Jim Kadonaga (Senior editor) and three reviewers, one of whom is a member of our Board of Reviewing Editors.

The reviewers have discussed the reviews with one another and the Reviewing editor has drafted this decision to help you prepare a revised submission.

Summary:

In this manuscript, the authors describe an archaeal TFIIE β homolog. Using the Sso protein, they show convincingly that it specifically dimerizes with TFEα. They show the Sso0944/TFEβ gene is essential for viability. They demonstrate that the β subunit, like the Pol III subunit RPC39, contains a 4Fe-4S domain. Mutagenesis and functional assays show that the FeS domain is important for dimerization of the α-β subunits. Finally, they demonstrate that the TFEα/β factor stimulates PIC stability, enhances DNA melting from a slightly pre melted promoter and stimulates transcription initiation. They conclude with an interesting speculation on the evolutionary history of TFE/TFIIE in LUCA, archaea and eukaryotes. These are important findings as they tie together the function of the archaeal and eukaryotic factors. Conceptually, these findings advance our understanding of the evolutionary relationship between transcription systems. The authors provide a clear and comprehensive description of a wide variety of experimental methods used in this study. This manuscript is of high quality and well suited for publication in *eLife* after successful revision.

Essential revisions:

1) The authors say that TFEβ has the potential to regulate transcription but they do not provide clear evidence for this. Please clarify or tone down statements.

2) Some of the pulldown assays in Figure 3 are confusing as the eluate sometimes contains a subunit marked as one of the TFE subunits that is not visible in the input protein. Please revise text or legends to make clear what is going on here.

3) In Figure 3 panel D a minimal TFEα/β assembly is shown after Ni affinity purification with a His-tagged b-subunit. However, the complex shown has much more of the a-subunit present. How can that be? This must be clarified.

4) In Figure 4 it is seen that some of the mutant TFEβ variants have a much higher absorption at 260 nm compared to 280 nm. This indicates contamination with nucleic acids. How could this influence the experiments?

5) The authors showed that deletions of the TFEα ZR removes binding to the clamp and a deletion of the TFEβ WH weakens PIC formation. However, in the third paragraph of the subsection headed “TFEα/β stabilizes the PIC and facilitates DNA melting” the authors describe that neither deletion perturbs promotor melting. From a kinetic point of view this is contradictory. However, in Figure 6 there might be actually less stimulation by these mutants. Please clarify.

6) In Figure 7, it is not obvious from my copy that TFE is stimulating abortive initiation (lanes 9,10). Please show quantitation of the products.

---

## [Author Response]

*1) The authors say that TFEβ has the potential to regulate transcription but they do not provide clear evidence for this. Please clarify or tone down statements*.

We do not provide direct evidence for regulation. Our data support the ‘potential’ of TFE to regulate transcription on a global scale. We have removed the strong statement from the Abstract. We have clarified and discussed the ‘potential’ of TFE to regulate transcription in a very careful and considered manner in the Discussion section: “…TFEα/β stimulates transcription of different genes to varying extent, dependent on the sequence of the initially melted region within the promoter. Thus, promoters that are strongly stimulated by TFEα/β could be downregulated in stationary phase, while expression of TFEα/β unresponsive (or only mildly stimulated) promoters would be less affected. We do not provide direct evidence for a regulatory role of TFEα/β, however, our results suggest that TFEβ has the potential to reprogram transcription in response to different growth phases.”

*2) Some of the pulldown assays in*
Figure 3
*are confusing as the eluate sometimes contains a subunit marked as one of the TFE subunits that is not visible in the input protein. Please revise text or legends to make clear what is going on here*.

The affinity column eluate fractions can contain tagged (as well as co-purifying) proteins at higher levels compared to the IP fraction since any efficient purification represents an enrichment. Overall, the heterologous expression of TFEα/β in *E. coli* results in higher TFEα levels compared to TFEβ (see Figure 3 ‘wt’ input versus eluate), and some of the TFEβ mutants are expressed at lower levels compared to wt TFEβ. This makes some of the TFE β variants difficult to discern in the input fraction, but since the co-purification strategy relies on tagged TFEβ, it is always present—and enriched—in the elution fractions. Slight variations of the apparent mobility of proteins are due to independent electrophoresis experiments collated in panel b. We systematically introduced red and blue triangles to identify the TFEα and β bands, respectively, and modified the figure legend as follows to clarify.

*3) In*
Figure 3
*panel D a minimal TFEα/β assembly is shown after Ni affinity purification with a His-tagged b-subunit. However, the complex shown has much more of the a-subunit present. How can that be? This must be clarified*.

The minimal TFEβ variant is very small (52 amino acids) and stains very poorly using Coomassie Blue as compared to TFEα (comprised of 114 amino acids). Since the complex was purified via the TFEβ his-tag the complex is stoichiometric despite the un-equivalent staining.

*4) In*
Figure 4
*it is seen that some of the mutant TFEβ variants have a much higher absorption at 260 nm compared to 280 nm. This indicates contamination with nucleic acids. How could this influence the experiments?*

The higher absorption at 260 nm relative to 280 nm in the 92S, 95S and 112S mutant variants (compared to wt nor 101S) is due to low levels of contaminants eluting slightly earlier than TFEα/β, and, as the reviewers indicate, possibly due to nucleic acids. The reasons that the contaminants’ contribution is higher in mutant than wt factors is simply due to the fact that their expression levels are considerably lower. In order to clarify this detail and allow a direct comparison, we have re-scaled the absorption y-axis of all elution profiles in Figure 4 to the same maximum (1,200 mU). In addition, we would like to emphasize that in all TFEα/β preparations (wt and mutant variants) the early fractions of the elution peak from the IMAC purification containing potential nucleic acid contaminants were discarded. IMAC purified TFEα/β preparations that were additionally purified via heparin-affinity chromatography were indistinguishable in all activity assays. As some of the mutant TFEα/β variants did not bind to heparin-sepharose, this purification step was omitted in experiments where mutant and wt TFEα/β were compared.

*5) The authors showed that deletions of the TFEα ZR removes binding to the clamp and a deletion of the TFEβ WH weakens PIC formation. However, in the third paragraph of the subsection headed “TFEα/β stabilizes the PIC and facilitates DNA melting” the authors describe that neither deletion perturbs promotor melting. From a kinetic point of view this is contradictory. However, in*
Figure 6
*there might be actually less stimulation by these mutants. Please clarify*.

In the complex formed between TFE and the recombinant RNAP clamp the latter is a minimal binding site. In this minimal complex the deletion of the TFEα ZR weakens the binding considerably. In contrast, in the DNA melting experiments a much larger interaction surface for TFE is presented—the entire RNAP—and in this context the ZR deletion has no phenotype presumably because additional contacts compensate for the weakened interaction. Robust EMSA signals require stable PICs because all components of the complex have to remain associated throughout the entire electrophoresis. EMSAs are nonequilibrium experiments, when PICs dissociate they will give rise to ‘fuzzy’ band shifts, or disappear altogether. This implies that EMSAs are very sensitive and ideally suited to test weakening effects of mutations, in this case TFE mutations such as the TFEβ WH deletion. This mutant, as well as the TFEα ZR deletion, barely stimulates PIC stability compared to the wt factor (Figure 6). The permanganate foot printing experiments have a completely different ‘window of opportunity’; as soon as the DNA strands are separated non-basepaired T-residues are modified (and subsequently cleaved). This in turn implies that the method is less sensitive to the effect of attenuated mutant variants, as our experiments demonstrate TFEα/β, α ΔZR/β, and α/β ΔWH all stimulate permanganate reactivity to a similar extent (Figure 6).

*6) In*
Figure 7*, it is not obvious from my copy that TFE is stimulating abortive initiation (lanes 9,10). Please show quantitation of the products*.

TFE reproducibly stimulates abortive transcription from both closed and preopened templates. To clarify this activity we have included the magnitude of the stimulation (2.3±0.5 and 1.6±0.1, respectively) in the manuscript text: “TFEα/β stimulates abortive transcription activity on closed promoter templates (2.3 ± 0.5 fold) as well as pre-opened (1.6 ± 0.1 fold) templates (Figure 7) while no stimulation was observed with TFEα (data not shown).”